# Variational Predictive Routing with Nested Subjective Timescales

**Alexey Zakharov**
Huawei Technologies
London, UK

**Qinghai Guo**
Huawei Technologies
Shenzhen, China

**Zafeirios Fountas** [*]
Huawei Technologies
London, UK

## Abstract

Discovery and learning of an underlying spatiotemporal hierarchy in sequential data is an important topic for machine learning. Despite this, little work has been done to explore hierarchical generative models that can flexibly adapt their layerwise representations in response to datasets with different temporal dynamics. Here, we present Variational Predictive Routing (VPR) – a neural probabilistic inference system that organizes latent representations of video features in a temporal hierarchy, based on their rates of change, thus modeling continuous data as a hierarchical renewal process. By employing an event detection mechanism that relies solely on the system's latent representations (without the need of a separate model), VPR is able to dynamically adjust its internal state following changes in the observed features, promoting an optimal organisation of representations across the levels of the model's latent hierarchy. Using several video datasets, we show that VPR is able to detect event boundaries, disentangle spatiotemporal features across its hierarchy, adapt to the dynamics of the data, and produce accurate time-agnostic rollouts of the future. Our approach integrates insights from neuroscience and introduces a framework with high potential for applications in model-based reinforcement learning, where flexible and informative state-space rollouts are of particular interest.

## 1 Introduction

A key theoretical benefit of hierarchically-structured latent models is the ability to learn spatial and temporal features that are characterised by an increasing level of abstraction in deeper layers. In most current implementations of such models, state transitions are realized using recurrent neural networks that operate at fixed rates determined by the timescale of the recorded data (Wichers et al., 2018; Castrejon et al., 2019; Kumar et al., 2019; Wu et al., 2021). Though quite successful, this approach neglects the role of the transition rates in the process of learning spatiotemporal representations across the model's latent hierarchy. Furthermore, it is computationally inefficient as more abstract features tend to remain static for longer periods than features represented in lower levels. This redundancy in state transitions introduces noise during both inference and future predictions, which is particularly detrimental to the learning of long-term dependencies (Bengio et al., 1994).

Having the ability to update the representation of these high-level features only when a change occurs in the environment can save significant computational resources and minimize the effect of accumulated noise in state transitions. Such an 'event-based' hierarchical model can also be beneficial for predicting future sequences (e.g. planning in model-based reinforcement learning) as the same future point in time can be reached with shorter and more accurate state rollouts (Pertsch et al., 2020; Zakharov et al., 2021). These benefits have recently prompted a number of proposed models that either perform transitions with temporal jumps of arbitrary length (Koutnik et al., 2014; Buesing et al., 2018; Gregor et al., 2018; Saxena et al., 2021), or aim to identify significant events (or key-frames) in sequential data and model the transitions between these events (Schmidhuber, 1991; Chung et al., 2017; Neitz et al., 2018; Jayaraman et al., 2018; Shang et al., 2019; Kipf et al., 2019; Kim et al., 2019; Pertsch et al., 2020; Zakharov et al., 2021).

The large variety of different event criteria defined in these studies demonstrates the lack of a widely established definition of this concept in this literature. For instance, important events are either

---

[*]Corresponding author: `zafeirios.fountas@huawei.com`

selected as the points in time that contain maximum information about a full video sequence (Pertsch et al., 2020), about the agent's actions (Shang et al., 2019), as the most predictable (Neitz et al., 2018; Jayaraman et al., 2018) or as the most surprising (Zakharov et al., 2021) points in time. A clearer picture can be drawn when viewing events from the perspective of cognitive psychology, where events are defined as segments of time "conceived by an observer to have a beginning and an end" (Zacks & Tversky, 2001). Neuroimaging evidence suggests that the human brain segments continuous experience hierarchically in order to infer the structure of events, with timescales that increase from milliseconds to minutes across a nested hierarchy in the cortex (Baldassano et al., 2017) while their boundaries tend to track changes in scene features and mental representations (Hard et al., 2006; Kurby & Zacks, 2008).

The processing of changes in the brain's hierarchical representations can be mathematically described using the framework of predictive coding (PC). According to PC, cortical representations of the world's state are constantly compared against new sensory information in multiple local inference loops that communicate via prediction error signals (Mumford, 1992; Rao & Ballard, 1999; Friston, 2010). In a contemporary view of this framework, called predictive routing, inference is realized via bottom-up propagation of sensory signals (rather than errors), which can be blocked by top-down predictions if considered uninformative (Bastos et al., 2020). This simple connection between salient representation changes and PC has led studies to propose plausible mechanisms for subjective experience of time (Roseboom et al., 2019) and the formation of episodic memories (Fountas et al., 2021) based on the detection of such changes.

Inspired by these insights, we propose a variational inference hierarchical model, termed variational predictive routing (VPR), that relies on a change detection mechanism to impose a nested temporal hierarchy on its latent representations. We implement this model in the domain of video using recurrent neural networks and amortised inference and show that our model is able to detect changes in dataset features even in early learning stages, regardless of whether they are expected or unexpected changes, with significant benefits for the learnt latent representations.

In summary, we make the following contributions. (a) We propose a mechanism that can detect the onset of both expected and unexpected events individually. (b) We show that event boundary discovery can rely on layerwise disentangled representations without the need of a dedicated learning mechanism. (c) We formulate a generative model (VPR) that integrates various insights from neuroscience and uses the detection mechanism to learn hierarchical spatiotemporal representations. (d) We propose a dataset based on 3D Shapes (Burgess & Kim, 2018) that allows the analysis of hierarchical latent spaces with nested temporal structure. (e) In different experiments and datasets, we demonstrate the ability of VPR to discover changes, be cost-effective, adjust representations to the dataset's temporal factors of variation and produce farsighted and diverse rollouts in the environment, all of which are key advantages for agent learning and planning.

## 2 RELATED WORK

**Hierarchical generative models** Our proposed model relates to the extensive body of literature on hierarchical generative models. Incorporation of the hierarchy into variational latent models has been used for improving the expressiveness of the variational distributions (Ranganath et al., 2016), modeling image (Rasmus et al., 2015; Bachman, 2016; Vahdat & Kautz, 2020) and speech data (Hsu et al., 2019). Notably, NVAE (Vahdat & Kautz, 2020) implement a stable fully-convolutional and hierarchical variational autoencoder (VAE) with the use of separate deterministic bottom-up and top-down information channels. NVAE incorporates similar architectural components to VPR but is designed to model stationary, as opposed to temporal, data. A more closely related hierarchical generative model is CW-VAE (Saxena et al., 2021). CW-VAE is designed to model temporal data using a hierarchy of latent variables that update over fixed time intervals. Unlike CW-VAE, our model employs an adaptive strategy in which update intervals rely on event detection.

Furthermore, hierarchical generative models are also subject to research in the area of PC. Deep neural PC was originally proposed in PredNet by Lotter et al. (2016), who implemented local inference loops that propagate bottom-up prediction errors using stacked layers of LSTMs. Although PredNet is able to predict complex naturalistic videos and to respond to visual illusions similar to the human visual system (Watanabe et al., 2018), it does not perform well in learning high-level latent representations (Rane et al., 2020). This led to its alternative implementations such as CortexNet (Canziani & Culurciello, 2017). More recently, a series of studies including Song et al. (2020); Millidge et al.

(2020) showed that deep PC can be implemented in a single computational graph and can, under some conditions, replace back-propagation with local learning updates. Although local inference loops in PC are suitable for identifying event boundaries, none of the above models involve the learning of temporal abstractions. An exception is the hybrid episodic memory model in Fountas et al. (2021), where temporal abstraction was introduced as a means to model human reports of subjective time durations. Finally, to our knowledge, there is currently no computational model of predictive routing (Bastos et al., 2020) – the contemporary view of PC that inspired our work.

**Hierarchical temporal structure**    A number of hierarchical approaches that do not rely on fixed intervals have also been put forward. Chung et al. (2017) proposed Hierarchical Multiscale RNNs and showed that a modified discrete-space LSTM can discover multiscale structure in data in an unsupervised manner. This method relies on a parameterized boundary detector and introduces additional operations to RNNs for controlling this detection, as well as copying and updating the current hidden state of the network. HM-RNNs have been combined with more advanced state transition functions (Mujika et al., 2017) and have been extended to a stochastic version using variational inference (Kim et al., 2019). A crucial difference between VPR and HM-RNNs is that, in our model, event boundaries are determined via a mechanism that does not involve learning. Instead, VPR detects changes in latent representations of each hierarchical layer independently, and enforces a nested temporal structure by blocking bottom-up propagation between the detected boundaries. As a result, event detection performance is high even in the very first stages of training while the emerging temporal structures change organically along with the latent space throughout training.

## 3    VARIATIONAL PREDICTIVE ROUTING

We introduce variational predictive routing (VPR) with nested subjective timescales – a hierarchical event-based generative model capable of learning disentangled spatiotemporal representations using video datasets. The representational power of VPR lends itself to several key components: (1) distinct pathways of information flow between VPR blocks, (2) selective bottom-up communication that is blocked if no change in the layerwise features is detected, and (3) the resultant subjective-timescale transition models that learn to predict feature changes optimally.

**Block architecture**    VPR consists of computational blocks that are stacked together in a hierarchy. Fig. 1 shows the insides of a single block with key variables and channels of information flow. Each block in layer $n$ consists of three deterministic variables $(x_\tau^n, c_\tau^n, d_\tau^n)$ that represent the three channels of communication between the blocks: bottom-up (encoding), top-down (decoding), and temporal (transitioning), respectively. These variables are used to parametrise a random variable $s_\tau^n$ that contains the learned representations in a given hierarchical level.

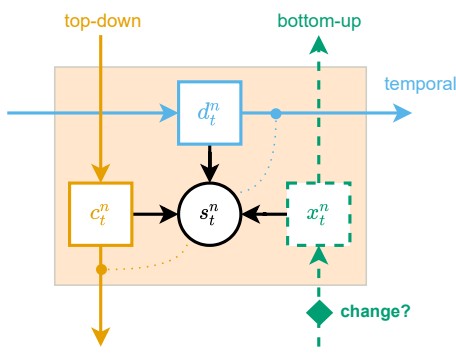

Figure 1: VPR block architecture. Distinct information channels are mediated using different deterministic variables: $d_\tau^n$ temporal, $c_\tau^n$ top-down, $x_\tau^n$ bottom-up. In turn, they are used to parametrise random variable $s_\tau^n$.

Communication between the blocks is a crucial component of the system. Top-down decoding from level $n+1$ to $n$ is realised by passing the latest context $c^{n+1}$ and sample from $s^{n+1}$ through a neural network to retrieve $c^n = f_{dec}(c^{n+1}, s^{n+1})$. Temporal transitioning is implemented with the use of a recurrent GRU model (Cho et al., 2014), such that $d_{\tau+1} = f_{tran}(s_\tau, d_\tau)$. Finally, bottom-up encoding of new observations iteratively computes layerwise observation variables, $x^{n+1} = f_{enc}(x^n)$. Fig. 2 shows a three-level VPR model unrolled over five timesteps, demonstrating how these communication channels interact over time.

**Subjective timescales**    As will be described next, each level $n$ updates its state only when an event boundary has been detected. This results in updates that can occur arbitrarily far apart in time – driven only by the changes *inferred* from the observations and dependent on the model's internal *representations*. For this reason, we use the term *subjective timescales* to refer to level-specific

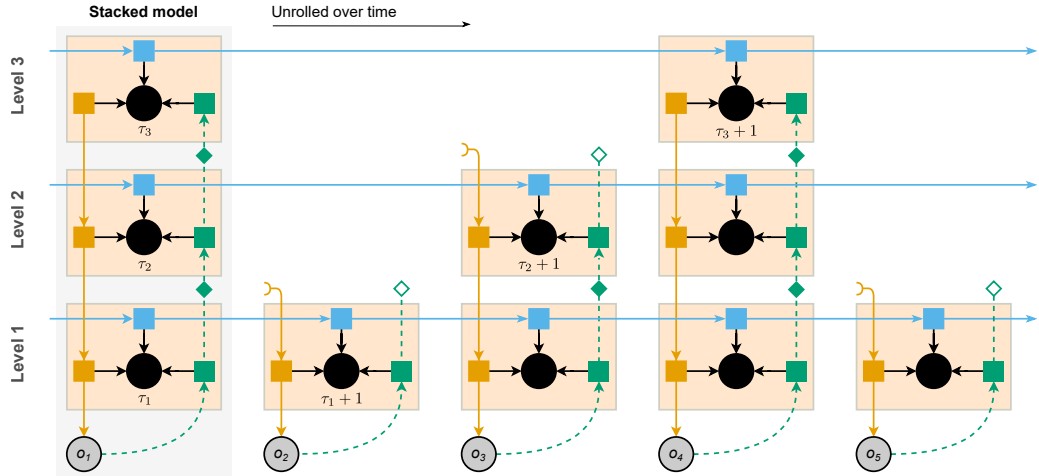

Figure 2: Example of a three-level VPR model unrolled over five timesteps. 🟦🟧🟩⚫ are block variables as demonstrated in Fig. 1. ⌐ indicates the latest top-down context from a level above. ◆ indicates that bottom-up encoding channel is open, while ◇ indicates that it is blocked.

succession of time and denote it by $\tau_n \in \{1, ..., T_n\}$, where $T_n$ is the total number of states produced in level $n$. The subjective time variable $\tau_n$ should not be confused with variable $t \in \{1, ..., T\}$ that denotes the objective timescale (i.e. normal succession of time). Lastly, we set $t = \tau_n$ for $n = 1$.

**Formal description**  We consider sequences of observations $\{o_1, ..., o_T\}$, modelled using a collection of latent variables $s_{1:T_{1:N}}^{1:N}$. The generative model of VPR can be written as a factorised distribution,

$$p(o_{1:T}, s_{1:T_{1:N}}^{1:N}) = \Big[ \prod_{\tau_1=1}^{T_1} p(o_{\tau_1}|s_{\tau_1}^{1:N}) \Big] \Big[ \prod_{n=1}^{N} \prod_{\tau_n=1}^{T_n} p(s_{\tau_n}^n|s_{<\tau_n}^n, s_{\tau>n}^{>n}) \Big], \tag{1}$$

where $s_{\tau_n}^{1:N}$ denotes all latest hierarchical states at timestep $\tau_n$, $p(s_{\tau_n}^n|s_{<\tau_n}^n, s_{\tau>n}^{>n})$ represents a prior distribution of the state $s_{\tau_n}^n$ conditioned on past states in level $n$ and all above states in the hierarchy $> n$, and $p(s_1^N) = \mathcal{N}(0, 1)$ is a diagonal Gaussian prior. To simplify the notation, we omit the subscript $n$ from $\tau_n$ and $T_n$ in the following sections and use $\tau$ and $T$ instead. Since the true posterior $p(s_{1:T}^{1:N}|o_{1:T})$ is intractable, we resort to approximate it using a parametrised approximate posterior distribution $q(s_{1:T}^{1:N}|o_{1:T}) = \prod_{\tau=1}^{T} \prod_{n=1}^{N} q_\phi(s_\tau^n|x_\tau^n, s_{<\tau}^n, s_\tau^{>n})$, where $x_\tau^n$ is the layerwise encoding of $o_\tau$ and $\phi$ are the parameters of the posterior model.

Notably, the proposed generative model is not Markovian in both temporal and top-down directions ($s_{<\tau}$ and $s^{>n}$) by virtue of the deterministic variables $d$ and $c$ that mediate block communications during the generative process. All of the model components are summarised as follows,

| | | | | | |
|---|---|---|---|---|---|
| Posterior model, | $q_\phi(s_\tau^n\|x_\tau^n, s_{<\tau}^n, s_\tau^{>n})$ | (2) | Encoder, | $x^{n+1} = f_{enc}^n(x^n)$ | (5) |
| Prior model, | $p_\theta(s_\tau^n\|s_{<\tau}^n, s_\tau^{>n})$ | (3) | Decoder, | $c^{n-1} = f_{dec}^n(s^n, c^n)$ | (6) |
| Transition model, | $d_{\tau+1}^n = f_{tran}^n(s_\tau^n, d_\tau^n)$ | (4) | Reconstruction, | $o_\tau = f_{rec}(c_\tau^0)$ | (7) |

Finally, to train VPR, we define a variational lower bound over $\ln p(o_{1:T})$, $\mathcal{L}_{ELBO}$,

$$\mathcal{L}_{ELBO} = \sum_{\tau=0}^{T} \mathbb{E}_{q(s_\tau^{1:N})}[\ln p(o_\tau|s_\tau^{1:N})] - \sum_{n=1}^{N} \sum_{\tau=1}^{T_n} \mathbb{E}_{q(s_\tau^{>n}, s_{<\tau}^n)}[D_{\mathrm{KL}}[\tilde{q}_\phi(s_\tau^n)||\tilde{p}_\theta(s_\tau^n)]]. \tag{8}$$

where $\tilde{q}_\phi(s_\tau^n) = q_\phi(s_\tau^n|o_\tau, s_{<\tau}^n, s_\tau^{>n})$ and $\tilde{p}_\theta(s_\tau^n) = p_\theta(s_\tau^n|s_{<\tau}^n, s_\tau^{>n})$.

**Event detection**  At the heart of VPR is a mechanism that is used for detecting predictable and unpredictable changes in the observable features over time, and thus determine the boundaries of events. Occurring at every level of the model, the event detection is used for controlling the structure

of the unrolled model over time by allowing (Fig. 2 ◆) or disallowing (Fig. 2 ◇) propagation of bottom-up information to the higher levels of the hierarchy.

Specifically, for some level $n$, the model retrieves the latest posterior belief state, $p_{st} = p(s_{\tau+1}^n | x_\tau^n, s_{<\tau}^n, s_\tau^{>n}) \equiv q_\phi(s_\tau^n | x_\tau^n, s_{<\tau}^n, s_\tau^{>n})$, which represents model's belief over a subset of observable features (represented in level $n$) at the latest observed timestep, $\tau$. Given a new observation, $x_{\tau+1}^n$, VPR then computes an updated posterior belief under the *static* assumption, $q_{st} = q_\phi(s_{\tau+1}^n | x_{\tau+1}^n, s_\tau^{>n}, s_{<\tau}^n)$. Notice that $q_{st}$ is conditioned on the same set of latent states, which implies that $p_{st} = q_{st}$ iff $x_\tau^n = x_{\tau+1}^n$. This means that the model's posterior belief over the inferred features will remain constant if the features have not changed between $\tau$ and $\tau + 1$. The model's update in the posterior belief is then measured using KL-divergence, $D_{st} = D_{KL}(q_{st}||p_{st})$. This key quantity can be seen as a measure of how much the relevant layerwise features have changed since the last encoded timestep $\tau$.

Next, VPR has two ways of identifying an event boundary. **Criterion E (*CE*)**: predictable events are detected with the use of the model's transition model that computes the next subjective timestep's belief state, $p_{ch} = p_\theta(s_{\tau+1}^n | s_\tau^n, s_{<\tau}^n, s_\tau^{>n})$. This prediction can be seen as the model's prior belief over the features it expects to observe next (the *change* assumption). Upon receiving the new information $x_{\tau+1}^n$, the model then calculates the posterior belief, $q_{ch} = q_\phi(s_{\tau+1}^n | x_{\tau+1}^n, s_\tau^n, s_{<\tau}^n, s_\tau^{>n})$. Similarly to the computations for the *static* assumption, the KL divergence between these two distributions is calculated, $D_{ch} = D_{KL}(q_{ch}||p_{ch})$. A predictable event is considered detected if $D_{KL}(q_{st}||p_{st}) > D_{KL}(q_{ch}||p_{ch})$. Satisfying this criterion indicates that the model's prediction produced a belief state more consistent with the new observation $x_{\tau+1}^n$, suggesting that it contains a predictable change in the features that are represented in the corresponding level of the hierarchy. **Criterion U (*CU*)**: events that are not or cannot be captured well by the transition models (i.e. unexpected changes) require an additional method for their detection. To this end, VPR keeps a moving average statistics of the latest $\tau_w$ timesteps over the observed values of $D_{st}$, and identifies an event when inequality $D_{st,\tau+1} > \gamma \sum_{k=\tau-\tau_w}^\tau D_{st,k}/\tau_w$ (where $\gamma$ is a threshold weight) is satisfied. As will be shown, this criterion proved to be crucial in the beginning of the training when the model's representations and layerwise transition models are not yet trained.

**Bottom-up communication**  Event detection serves two primary functions in our model. First, detecting layerwise events in a sequence of observations is used for triggering an update on a block's state (Fig. 2 ◆) by inferring its new posterior state. Matching block updates with detectable changes in layerwise features (event boundaries) prompts VPR to represent spatiotemporal features in levels that most closely mimic their rate of change over time. Similarly, learning to transition between states *only* when they signify a change in the features of the data allows VPR to make time-agnostic (or jumpy) transitions – from one event boundary to another. Second, the detection mechanism is used for blocking bottom-up communication (Fig. 2 ◇) and thus stopping the propagation of new information to the deeper levels of the hierarchy. This encourages the model to better organise its spatiotemporal representations by enforcing a temporal hierarchy onto its generative process.

Practically, when a bottom-up information channel is blocked at level $n$, the variables of the model in levels $\geq n$ will remain unchanged. By virtue of the hierarchy, this similarly means that the top-down information provided to level $n - 1$ at the next timestep will stay constant (Fig. 2 ↰). As a result, the blockage mechanism encourages the model to represent slower changing features in the higher levels of the hierarchy.

Importantly, there is no constraint on when and how often a state update can occur over time. In other words, VPR is time-agnostic and is designed to model the changing parts of sequences with temporal hierarchy, irrespective of how slow or fast their rates are. This leads to the model's ability to dynamically adjust to the datasets with different temporal structures and to learn more optimal, disentangled, and hierarchically-structured representations. Fig. 3 shows the ability of VPR to adjust the rate at which blockage occurs to datasets with slower and faster temporal dynamics.

## 4 EXPERIMENTS

Using a variety of datasets and different instances of VPR, we evaluate its performance in the following areas: (a) unsupervised event boundary discovery, (b) generation of temporally-disentangled hierarchical representations, (c) adaptation to the temporal dynamics of incoming data, and (d) future event prediction. We employ the following datasets and their corresponding VPR model instances:

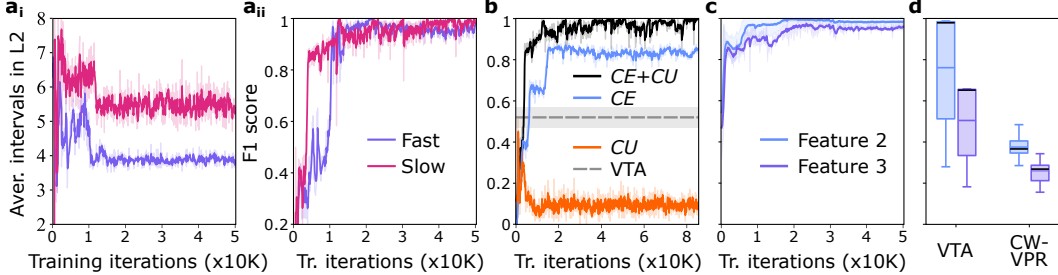

Figure 3: Performance of VPR's event detection system using Moving Ball and 3DSD. ($\mathbf{a}_i$) VPR adapts its rate of state updates to the temporal dynamics of the Moving Ball dataset, ($\mathbf{a}_{ii}$) while maintaining the same accuracy of event detection. (**b**) F1 score of event detection using different sets of criteria in Moving Ball, with a comparison against VTA (Kim et al., 2019). (**c**) F1 score of levels 2 and 3 in detecting changes in their corresponding features using 3DSD. (**d**) Comparison against baseline methods using 3DSD. Shaded regions indicate one standard deviation over 5 seeds.

**Synthetic dataset** is implemented to validate our approach for event boundary discovery and analyse the roles of the chosen event criteria. The dataset consists of 1D sequences generated using a semi-Markov process with random value jumps every 10 steps. The VPR instance employed is a minimal version, where $N = 1$, the transition model $p_{ch}(s_{\tau+1}|s_\tau)$ is memoryless and the posterior state calculation is replaced by a direct map $q_{st}(s_\tau^1) = q_{ch}(s_\tau^1) = o_\tau + c$, where $c$ is Gaussian noise.

**Moving Ball** consists of a coloured ball that travels in straight lines and bounces off walls within the boundaries of an image. Importantly, the ball's colour changes either upon a wall bounce or at a random time with probability $p = 0.1$. As there are two dynamic features – position and colour of a ball – we employ VPR with $N = 2$ (two levels) and measure the accuracy of the model to detect and represent changes in the ball's colour in level 2 of the hierarchy.

**3D Shapes Dynamic (3DSD)** is a dynamic extension to the 3D Shapes dataset (Burgess & Kim, 2018) – a large image collection of shapes generated using 6 underlying factors of variation. By manipulating a subset of these factors, we produce sequences of images where different factors vary over time with different periodicity. Specifically, we impose a temporal hierarchy onto three factors – colours of objects, walls, and floor (from slow to fast, respectively). We test the ability of VPR with $N = 3$ to detect and represent the changes in these features in levels 2 and 3 of the hierarchy.

**Miniworld Maze.** To evaluate the behaviour of VPR in a more perceptually challenging setting, we use a 3D environment Gym-Miniworld (Chevalier-Boisvert, 2018). For this, we designed a maze environment that consists of a sequence of connected hallways – each with a different colour and of a random length – and collect sequecnes of agent observations when traversing these hallways.

To validate event boundary discovery in VPR, we compare against the temporal abstraction model that employs a parametrised boundary detection system, VTA (Kim et al., 2019). Further, to emphasise the benefit of flexible timescales we demonstrate the event detection accuracy of a CW-VPR – a version of VPR where latent transitions occur at fixed rates, analogous to CW-VAE (Saxena et al., 2021). Finally, we compare VPR's future event prediction accuracy against a model for long-term video prediction, CW-VAE (Saxena et al., 2021).

### 4.1 UNSUPERVISED EVENT DISCOVERY

VPR's event detection mechanism proves to be effective across three different datasets. Specifically, it quickly achieves high F1 score in the Synthetic (Fig. 4), Moving ball (Fig. 3b; F1 = $0.97 \pm 0.03$), and 3DSD (Fig. 3c; F1 = $0.97 \pm 0.02$) datasets. Furthermore, we observe a significantly better performance compared to the baseline models VTA (see Fig. 3b; Moving ball, Fig. 3d; 3DSD, Fig.4d; Synthetic) and CW-VPR (see Fig. 3d; 3DSD).

**Detection criteria analysis** As mentioned, the detection mechanism employs two criteria for identifying expected (*CE*) and unexpected (*CU*) events. Using the Synthetic and Moving Ball datasets, we show that the optimal performance is only achieved when both of these criteria are used in the decision-making. First, we observe that the role of *CU* is particularly crucial in the early stages of training, in order to provide initial supervision signal for the untrained transition model. Fig. 4a shows the F1 score of the *CE* criterion at detecting event boundaries with (blue curve) and

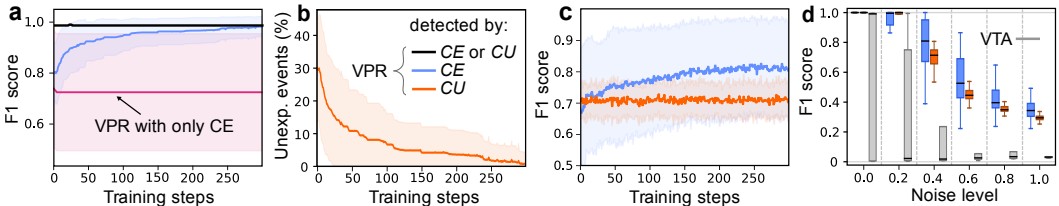

Figure 4: Event discovery performance in synthetic data. Posterior noise is set to 0.0 in (a-b) and 0.4 in (c). Shaded regions indicate one standard deviation over 5 seeds.

without (pink curve) the *CU* criterion enabled at the same time. It is evident that the transition model cannot improve performance with a disabled *CU*. In contrast, expected event recognition with both *CE* and *CU* quickly converges to optimality after some training. Notably, the better *CE* becomes in detecting events, the less number of events are classified as unexpected (Fig. 4b). Furthermore, in the later stages of learning, the roles of the two criteria switch, as *CE* becomes the driving force for improving the model's performance. Since detection with *CU* relies on a non-parametric threshold, its performance has a theoretical limit and reduces substantially for high levels of posterior noise (Fig. 4c-d). In contrast, *CE* continues to improve beyond *CU*'s supervision signal, being significantly more robust to noise. In addition, training VPR on the Moving Ball dataset using the different sets of detection criteria demonstrates similar behaviour. Fig. 3b shows that employing both *CE* and *CU* results in the significantly better and faster convergence of the detection F1 score.

## 4.2 HIERARCHICAL TEMPORAL DISENTANGLEMENT

VPR employs its hierarchical structure to disentangle the temporal features extracted from the dataset. By matching the rates of event boundaries with the rates of level updates, the model learns to represent particular temporal attributes of the dataset in the appropriate layers of the hierarchy. We analyse the property of hierarchical disentanglement by (1) producing rollouts using separate levels of the model, (2) generating random samples from different levels, and (3) analysing factor variability of the samples as a measure of disentanglement.

Each layer of VPR is equipped with an event detection mechanism that guides the model to distribute representations of temporal features across the hierarchy and learn jumpy predictions between event boundaries. Fig. 5 shows layerwise rollouts of VPR in the Moving Ball and 3DSD datasets. For Moving Ball in Fig. 5a, VPR learns to represent the ball's position and colour in the two separate levels (L1 and L2, respectively). L1 rollout accurately predicts positional changes of the ball, while

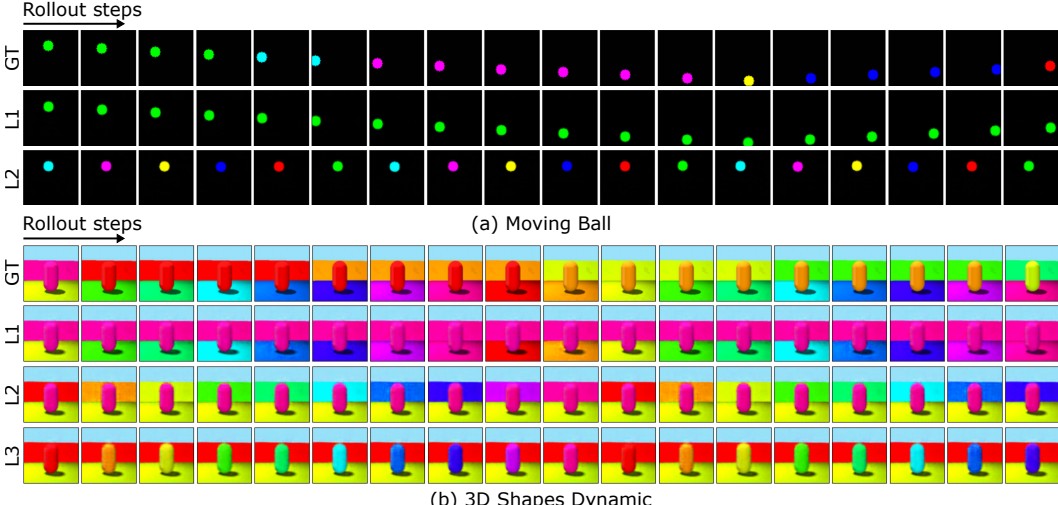

Figure 5: Layerwise rollouts using VPR. *GT* denotes ground-truth sequence, *L1* rollouts made using level 1, and so on. To produce layerwise rollouts, the model predicts the next state $s_{\tau+1}^n$ in the relevant level $n$ and decodes under fixed states in all other levels. The produced rollouts illustrate model's ability to learn disentangled representations and produce accurate and feature-specific jumpy rollouts.

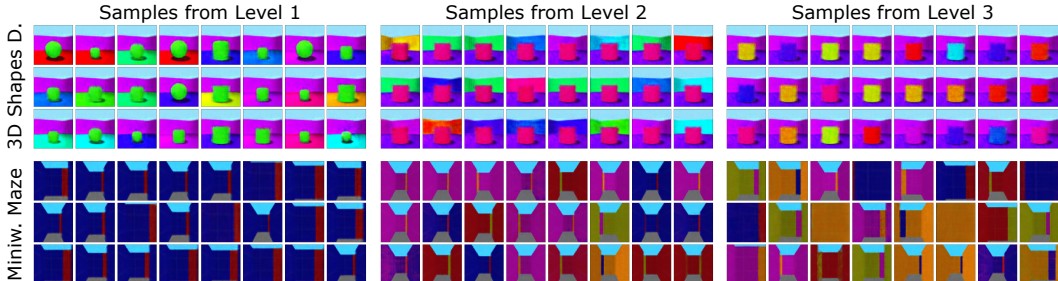

Figure 6: Random samples taken from the different levels of VPR. The model generates diverse images with respect to the spatiotemporal features represented in the sampled level, while keeping all other features fixed.

keeping the colour constant. At the same time, L2 rollout produces jumpy predictions of the ball colour under constant position. Similar behaviour can be observed in 3DSD, using three layers.

VPR's representational power can be further dissected by taking random samples from the different levels of the hierarchy. Fig. 6 shows 3DSD and Miniworld Maze reconstructions of VPR conditioned on sampling only one of the levels and fixing all others. In line with the produced rollouts, 3D Shapes samples indicate that L1 contains representations of the floor colour and object shape, L2 of wall colour and angle, and L3 of object colour. For the Miniworld Maze samples, we similarly observe that L1 represents the short-term feature of the agent's position in a room, L2 encodes the colour of the walls, while L3 provides the wider context of the agent's location in a maze.

To quantify temporal feature disentanglement, we measure the average entropy in the distribution of each of the features associated with the reconstructed samples, $H_v = -\frac{1}{M} \sum_m^M \sum_i^I p(v_i) \log p(v_i)$, where $I = 32$ is the number of samples per trial, $M = 100$ is the number of sampling trials, and $v$ is a factor of variation extracted from each reconstructed image using a pre-trained extractor model. Factor $v$ will be associated with higher average entropy if the layerwise random samples produce more uniform distributions of this factor across the reconstructed images. Fig. 7 shows that, for each level, the average entropy is high only for a temporal feature that ranks the same in the order of the dataset's temporal hierarchy (factor 1: fastest, factor 3: slowest). This implies that VPR distributes representations of features in a way that preserves the underlying temporal hierarchy of the dataset.

We contrast this result with instances of VPR that perform block updates over fixed intervals (analogous to CW-VAE (Saxena et al., 2021)) in Fig. 7. It can be seen that different intervals result in the different representational properties of the model, suggesting the difficulty of selecting the appropriate interval values. Furthermore, fixed intervals unnecessarily bound the unrolled structure of the model, meaning that features that occur over arbitrary time intervals are likely to not be abstracted to the higher levels. For instance, as shown in the Appendix Fig. 12, a fixed-interval VPR model cannot represent the ball colour in level 2 entirely, in contrast to VPR with subjective timescales.

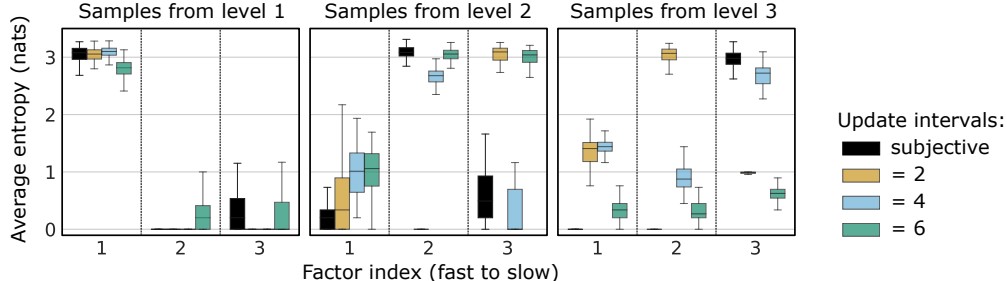

Figure 7: Feature disentanglement in 3DSD. VPR with subjective timescales finds the most appropriate distribution of representations, in contrast to fixed-interval models.

### 4.3 ADAPTATION TO TEMPORAL DYNAMICS

Since VPR is an event-based system, it relies solely on the underlying change dynamics of a dataset – layerwise computations are only initiated when an event boundary has been detected – and can thus

adapt to datasets with different temporal dynamics. To test this property, we instantiate two versions of the Moving Ball dataset – with fast and slow ball movements. Fig. 3a compares the average time intervals between block updates (in level 2) of VPR trained using these two datasets. It can be seen that VPR performs about 30% fewer block updates under slow ball movement compared to VPR under fast ball movement, while maintaining high F1 score of event detection in both cases. We also validate that both models produce the same hierarchical organisation of features reported in Section 4.2. This event-driven property implies significant computational savings, promoting a more optimal usage of computational resources with respect to the temporal dynamics of different datasets – which will be even more significant in VPR instances with a larger number of hierarchical levels.

### 4.4 FUTURE EVENT PREDICTION

The ability of VPR to learn jumpy and feature-specific transitions in its hierarchical levels has beneficial implications on the accuracy with which long-term event sequences can be predicted. To test this, we perform jumpy layerwise rollouts of length 200 for the Moving Ball and 3DSD datasets and evaluate them against the ground-truth event sequences – colour changes in the Moving Ball (VPR level 2), and wall and object colour changes in the 3DSD (VPR levels 2 and 3). Here we compare VPR against a powerful video prediction model, CW-VAE (Saxena et al., 2021), which produces rollouts over the physical timescale and thus requires longer rollouts to reach an equivalent of VPR's jumpy predictions. For instance, predicting 200 events of object colour changes in the 3DSD corresponds to a physical rollout of 1600 steps. For the 3DSD, we find that VPR make more accurate event predictions of both wall (acc. $0.99 \pm 0.017$) and object (acc. $0.90 \pm 0.04$) colour changes than CW-VAE (acc. $0.92 \pm 0.05$ and $0.71 \pm 0.02$, respectively). For the Moving Ball, we find that VPR significantly outperforms CW-VAE at predicting changes in the ball colour, reaching a perfect accuracy of $1.0 \pm 0.0$ against only $0.47 \pm 0.09$ by CW-VAE.

## 5 DISCUSSION

**Event duration estimation** The presented model in its current form is not able to generate far-sighted rollouts over the physical timescale of the data. This process requires an additional method to estimate the number of transitions in each layer $n$, before layer $n + 1$ performs a transition. This can be implemented with a neural network that estimates the probability $p(\text{change}|s_n, s_{n+1})$, trained in a self-supervised manner, using the criteria *CE* and *CU*. Analogous estimation has been employed in previous studies as a parametrised event boundary detector (Chung et al., 2017; Mujika et al., 2017; Kim et al., 2019). However, unlike these proposals, this estimation is not necessary for any of the results presented here, nor for potential applications such as agent planning or solving downstream tasks. To highlight this important difference, we chose to not explore event duration estimation here.

**Expected and unexpected events** The criterion for event boundary detection used to identify expected changes (*CE*) can be viewed as a model selection method based on the shortest belief update, given the prior assumption that a change has either occurred or not. If $q_{\text{ch}} \equiv q_{\text{st}}$, then the inequality *CE* is equivalent to maximizing the likelihood of the latent state given new observations. Indeed, we can see in Appendix Fig. 16 that the absolute difference between the entropies $H_{\text{ch}}(s^n)$ and $H_{\text{st}}(s^n)$ is negligible compared to the difference between the corresponding relative entropies, indicating that the *CE* criterion is predominantly governed by the latter difference. Furthermore, although detecting the onset of anticipated events is crucial for constructing hierarchical temporal latent spaces, these events do not denote salience, unless *CU* is also satisfied. Visual salience (in the form of attention in humans) has been often directly linked to the value of $D_{KL}(q||p)$, i.e. Bayesian surprise, with empirical evidence provided by Itti & Baldi (2006). Thresholding this quantity has been proposed as a model of attention to perceptual changes, which determines which changes can be deemed salient (Roseboom et al., 2019; Fountas et al., 2021), while the accumulation of such changes is shown to correlate well with cortical activity when humans perform duration judgments (Sherman et al., 2020). Finally, the distinction between expected and salient events, captured by *CE* and *CU* here, has been extensively discussed by Yu & Dayan (2005), who proposed a connection with two neuromodulators and an inextricably intertwined role in attention. Modeling hierarchical attention in VPR using *CE* and *CU* constitutes an interesting avenue for future work and yet another potential example of how cognitive neuroscience and machine learning can facilitate each other's progress.

ACKNOWLEDGMENTS

The authors would like to thank Yansong Chua for his valuable contributions and comments on the early versions of the model presented in the current manuscript.

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

## A  EVENT DETECTION MECHANISM

### A.1  ARCHITECTURE AND PARAMETERS

Fig. 8 shows the insides of the event detection mechanism implemented at each level of the model's hierarchy. Given the current timestep $\tau + 1$, the mechanism at level $n$ is triggered if the bottom-up communication has not been blocked by the level below, $n - 1$. Upon receiving new bottom-up information $x_{\tau+1}^n$, the model proceeds in evaluating four key variables. First, the latest posterior is assigned to be the new prior under the *static* assumption, $p_{st} = p(s_{\tau+1}^n | x_\tau^n, d_\tau^n, c_\tau^n) \leftarrow q_\phi(s_\tau^n | x_\tau^n, d_\tau^n, c_\tau^n)$. For clarity, we use the deterministic variables $d_\tau^n$ and $c_\tau^n$ to represent $s_{<\tau}^n$ and $s_\tau^{>n}$, respectively. Second, the new posterior under the *static* assumption is computed using the deterministic variables of the latest block, $q_{st} = q_\phi(s_{\tau+1}^n | x_{\tau+1}^n, d_\tau^n, c_\tau^n)$. As explained in the main body, we then compute the KL-divergence between the two states, $D_{st} = D_{KL}(q_{st} || p_{st})$. Third, we trigger the transition model to predict the next temporal context, $d_{\tau+1}^n$, in order to produce the prior under the *change* assumption of the model, $p_{ch} = p_\theta(s_{\tau+1}^n | d_{\tau+1}^n, c_\tau^n)$. Lastly, the posterior under the *change* assumption can also be computed using the new bottom-up encoding, $q_{ch} = q_\theta(s_{\tau+1}^n | x_{\tau+1}^n, d_{\tau+1}^n, c_\tau^n)$. As with the static assumption, we calculate the KL-divergence between the prior and posterior states, $D_{ch} = D_{KL}(q_{ch} || p_{ch})$.

Practically, prior state $p_{ch}$ is computed only once, after which it is stored for subsequent comparisons until the event criteria are satisfied and the block is updated.

Additionally, criterion $CU$ ($D_{st,\tau+1} > \gamma \sum_{k=\tau-\tau_w}^{\tau} D_{st,k} / \tau_w$) involves two hyperparameters: $\tau_w$ is the length of a sliding window used for calculating the moving average, and $\gamma$ is the threshold factor that multiplies the value of the moving average. In running the experiments, we found that the optimal values are $\gamma = 1.1$ and $\tau_w = 100$. These values also proved to be robust for use across different datasets, as we kept their values constant for all of the reported experiments.

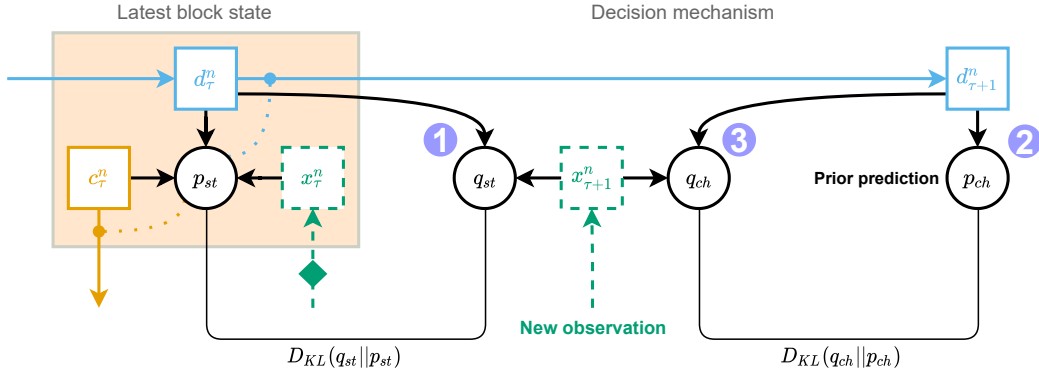

Figure 8: Event detection mechanism relies on the computation of key variables shown in the figure. Given the latest updated state of a VPR block at level $n$ and timestep $\tau$ denoted as $p_{st}$ (and the corresponding deterministic states $d_\tau^n$ and $c_\tau^n$), the model receives a new observation $x_{\tau+1}^n$ at some timestep $\tau + 1$. ① This new observation is used to compute the model's posterior belief, $q_{st}$, using the latest deterministic variables of the block. ② Concurrently, VPR makes a prediction using its generative model, producing $p_{ch}$ (and the corresponding $d_{\tau+1}^n$) representing the model's prior belief about the features at the next subjective timestep. ③ Lastly, VPR produces a posterior belief state, $q_{ch}$, under the updated temporal context variable $d_{\tau+1}^n$.

### A.2  DECISION METRICS

We can visualise the values calculated as part of the decision-making process in the event detection mechanism in Figure 9. As described in Section 4.1, the $CU$ criterion acts as the initial supervision signal, which subsequently results in the rapidly improving transition model and thus the $CE$-based detection. Figure 9 shows two examples of the computed values over the length of an observation sequence at the early (left) and later (right) stages of training. It can be observed that at only 500 training iterations the decision-making is primarily driven by the $CU$ criterion. At 18500 iterations, their roles tend to switch, as the $CE$ criterion becomes significantly more accurate at detecting events.

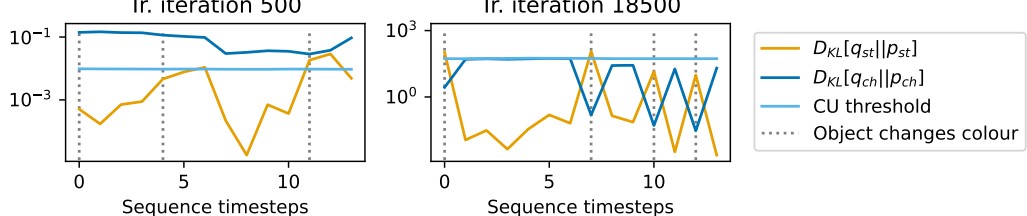

Figure 9: Key KL-divergence values computed using the level 2 detection mechanism at different stages of the training process using the Moving Ball dataset. Left-hand side graph shows the early stages of the training (training iteration 500), while right-hand side the later stages (training iteration 18500). In line with the described detection criteria in Section 3, if one of the blue lines falls below the orange line, an event is considered to be detected. As such, it can be seen that the decision-making is dominated by the *CU* criterion (light blue) in the early stages (left). On the other hand, as representations mature and the transition model learns, the *CE* criterion (dark blue) begins to dominate the detection process (right).

## B    MODEL ARCHITECTURE AND TRAINING

The model consists of several components implemented using neural networks:

- **Bottom-up**. Encoder is a combination of (a) a convolutional neural network that embeds high-dimensional image data into a lower-dimensional representation, (b) stacked fully-connected networks with residual connections $f_{enc}^n$, and (c) fully-connected networks in each layer that compress layerwise observation embeddings, $x^n$, prior to being passed into a posterior model.

- **Top-down**. Decoder is a combination of (a) a transpose convolutional neural network, $f_{rec}$, for reconstructing images using top-down information $c^0$ and (b) stacked fully-connected networks with residual connections $f_{dec}^n$.

- **Temporal**. Layerwise transition models are reccurrent GRU models (Cho et al., 2014) with hidden states of size $|d_\tau^n| = 200$ and an additional fully-connected network with $|s_\tau^n|$ neurons.

- **Prior and posterior models.** These models are implemented using four fully-connected networks and parametrise a diagonal Gaussian, thus outputting a vector of size $|s_\tau^n| \cdot 2$.

The convolutional components of the encoder and decoder are analogous to those used in Ha & Schmidhuber (2018). Fully-connected top-down and bottom-up components are all made out of the same building block, which consists of four fully-connected layers with a residual connection at the output (e.g. $x^{n+1} = f_{enc}^n(x^n) + 0.1 \cdot x^n$) and Leaky ReLU activations (Maas, 2013). The number of neurons is kept the same throughout and is equal to the dimensionality of the block's input (e.g. $|x_\tau^n|$). Component (c) of the bottom-up model, as well as the posterior and prior models, also consist of four fully-connected layers but with no residual connections.

We use the same VPR architecture for the Moving Ball and 3DSD datasets. Specifically, the latent states are of size $|s_t^n| = 20$, while the temporal, top-down, and bottom-up deterministic variables are set to be $|x_\tau^n| = |c_\tau^n| = |d_\tau^n| = 200$. In Moving Ball and 3DSD, observations $o_t \in \mathbb{R}^{64 \times 64 \times 3}$, so we set the model's input layer to have the same shape. For the Bouncing Balls dataset (see section C.3), we increase the capacity of the model, such that $|x_\tau^n| = 1024$ and $|s_t^n| = 60$.

For training, we use Adam optimizer (Kingma & Ba, 2015) with a learning rate of $0.0005$ and a cosine decay to $0.00005$ over a period of $15,000$ iterations. We employ linear annealing of the KL coefficient from $0$ to $1$ over the first $3000$ iterations. Although we also used KL balancing (Vahdat & Kautz, 2020) in the models presented in this paper, we find that it does not have a significant effect on their resultant properties. Further, we use binary cross entropy reconstruction loss and sequences of length 15 for the Moving Ball and Bouncing Balls datasets, and mean squared error loss and sequences of length 50 for 3DSD and Miniworld Maze datasets. Batch size of 32 is used for all datasets.

## B.1 Pseudocode of event detection

---

**Algorithm 1:** Event detection and inference in VPR

---

**for** $video = 1$ **to** $\infty$ **do**
  **for** $\tau = 1$ **to** *max length* **do**
    Retrieve a new observation, $o_\tau$
    Initialise an empty decision mask, list M

    `// Event detection (bottom-up)`
    **for** $n = 1$ **to** $N$ **do**
      Compute layerwise encoding via $f_{enc}^n, x_\tau^n$

      `// Level 1 decision is always True`
      **if** $n = 1$ **then**
        M.append(True)
        **continue**

      `// If propagation is blocked`
      **if** $M[n-1]$ *is False* **then**
        M.append(False)
        **continue**

      Compute variables $q_{ch}, q_{st}, p_{ch}, p_{st}$ of level $n$

      `// CE or CU criterion`
      **if** $D_{KL}(q_{st}||p_{st}) > D_{KL}(q_{ch}||p_{ch})$ *or* $D_{st,\tau} > \gamma \sum_{k=\tau-1-\tau_w}^{\tau-1} D_{st,k}/\tau_w$ **then**
        M.append(True)
      **else**
        M.append(False)

    `// Inference (top-down)`
    **for** $n = N$ **to** *0* **do**

      **if** $M[n]$ *is False* **then**
        **continue**

      **if** $n = 0$ **then**
        Compute $c_\tau^0$ using decoder, $c_\tau^0 \leftarrow f_{dec}^0(s_\tau^1, c_\tau^1)$
        Compute reconstruction image, $o_\tau \leftarrow f_{rec}(c_\tau^0)$
        **break**

      Compute $c_\tau^n$ using decoder, $c_\tau^n \leftarrow f_{dec}^n(s_\tau^{n+1}, c_\tau^{n+1})$
      Infer the new posterior state $s_\tau^n$ using the posterior model, $q_\phi(s_\tau^n | x_\tau^n, s_{<\tau}^n, s_\tau^{>n})$

---

# C Additional results

## C.1 Exploring information pathways

VPR is equipped with two distinct generative information pathways – temporal and top-down – that give the model flexibility in generating diverse sequences. We investigate this property further by performing layerwise rollouts using level $n = n_r$ of the model and setting $d_\tau^{<n_r} = 0$. We call the rolled out level a *target* level. Setting the temporal context variable to zero for all the levels below the target level effectively means that the model would have no temporal information about previously inferred states in those levels, $s_{<\tau}^{<n_r}$.

Figures 10-11 show layerwise rollouts under empty temporal priors (in all levels below) using the Moving Ball and 3DSD datasets. Additionally, we differentiate between rollouts with and without sampling – (a) rollouts with sampling are decoded by taking random samples from all states below the target level, $s_\tau^{<n_r}$; (b) rollouts without sampling are decoded using the means of the states' Gaussian distributions $s_\tau^{<n_r}$, instead.

**Moving Ball** Temporal information carried in level 1 relates to the ball's position and direction of travel. After depriving the model of this information and performing a rollout using level 2, we observe a distinct behaviour, in which VPR correctly predicts the changes in the ball's colour while randomly assigning it a position within an image (Fig. 10a). When no sampling is done, the position remains constant in the middle of the box (Fig. 10b).

**3DSD** As was seen in the main body of the paper, VPR learns to represent different features of the 3D Shapes dataset in the different levels of its hierarchy. As a result, performing layerwise rollouts under empty temporal priors in the levels below a rolled out level produces interesting properties. Rolling out level 3 while sampling during decoding (Fig. 11) creates a sequence in which the changes in the object colour (a feature that is represented in level 3) are correctly predicted over the steps of the rollout; however, all other features – including object shape, angle, wall and floor colours – are decoded at random. Given that these features are represented in levels 1-2 and that no temporal information was passed to these levels, the model predicts high uncertainty for latent variables $s_\tau^{<n_r}$ and thus samples the represented factors at random. Level 2 rollouts show the same behaviour – while the wall colour and object shape are correctly predicted over the timesteps of the rollout, the floor colour (represented in level 1) is sampled at random. These rollouts once again demonstrate VPR's disentanglement properties, as well as its versatility in generating sequences of quality reconstructions even when information from one of the channels is blocked.

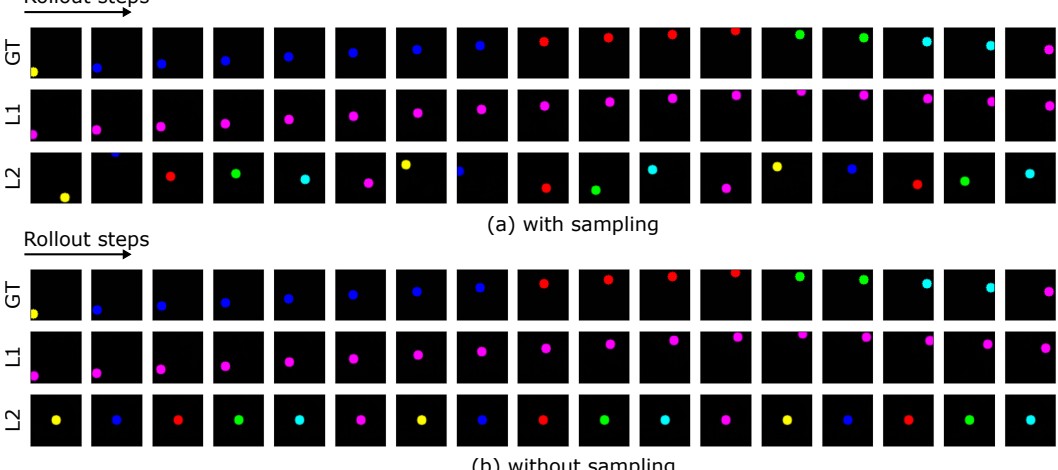

Figure 10: Layerwise rollouts under empty temporal priors in the Moving Ball dataset. *GT* denotes the ground-truth sequence, *L1* level 1 rollout, and so on. (a) Decoding performed while sampling at all levels below the target level; (b) decoding is done using the means of the Gaussians for $s_t^{<n_r}$.

### C.2 Fixed-interval models comparison

One of the advantages of VPR with subjective timescales (VPR-ST) over its equivalent fixed-interval models is the ability to abstract temporal features to the deeper levels of its hierarchy even when changes in these features occur over arbitrary number of timesteps. The Moving Ball dataset allows us to analyse this property of VPR, since the ball's colour does not change over fixed time intervals.

To test this, we train four different fixed-interval VPR models that update their level 2 state every 2, 4, 6, and 8 timesteps. Figure 12 shows layerwise rollouts of these models, which are analogous to Figure 5 where we analysed subjective-timescale VPR. As can be seen, the model struggles to abstract the colour of the object entirely to level 2, in contrast to VPR-ST. Because the colour of the object does not change with fixed periodicity, fixed-interval models do not seem to be suitable for learning temporally disentangled features in such datasets.

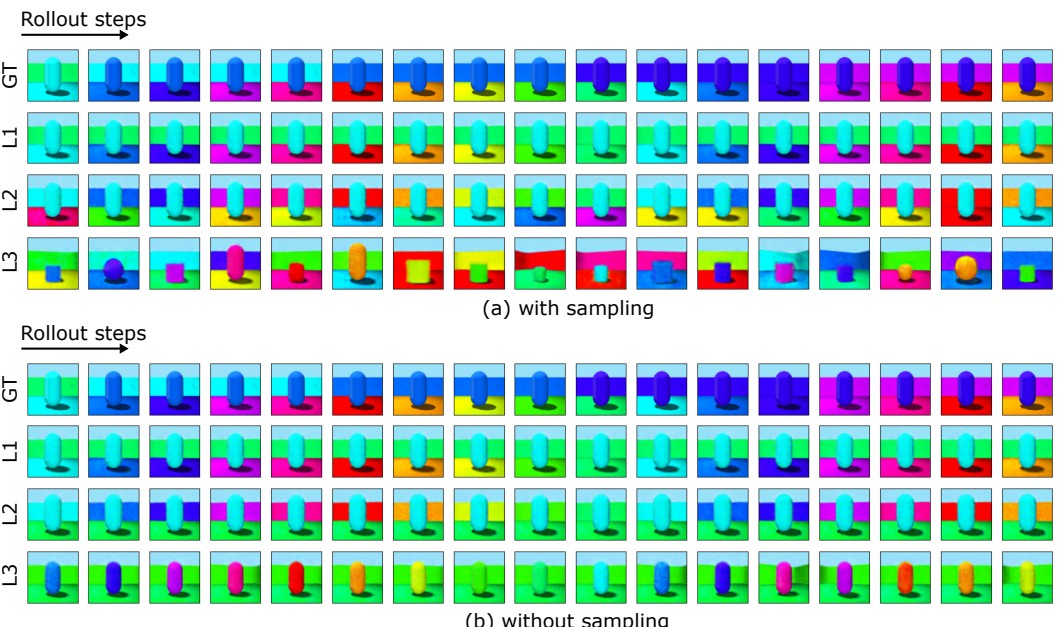

Figure 11: Layerwise rollouts under empty temporal priors in the 3DSD dataset. *GT* denotes the ground-truth sequence, *L1* level 1 rollout, and so on. (a) Decoding performed while sampling at all levels below the target level; (b) decoding is done using the means of the Gaussians for $s_t^{<n_r}$.

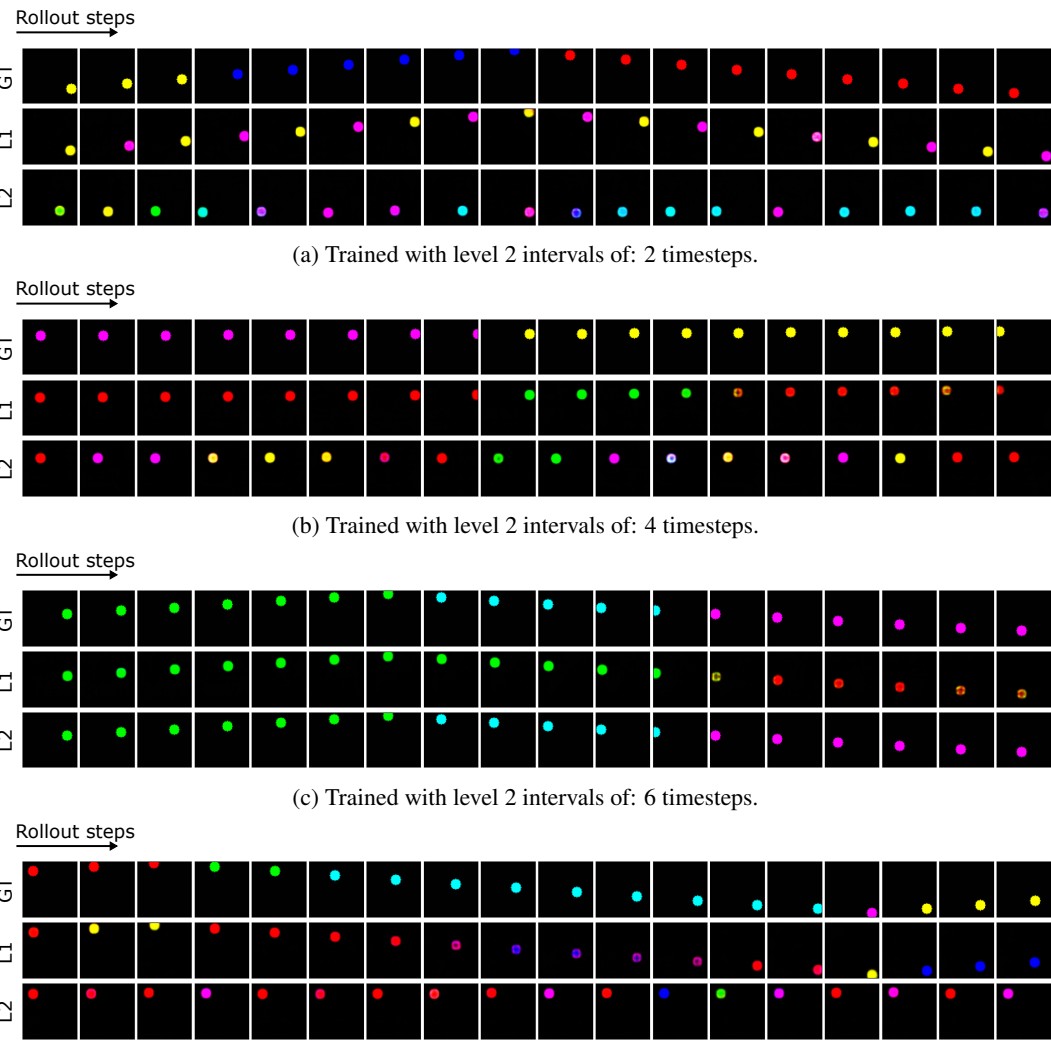

Figure 12: Layerwise rollouts using fixed-interval VPR models with manually assigned intervals of 2, 4, 6, and 8. It can be observed that the ball colour does not get entirely represented in level 2 (L2) of the model, in contrast to the VPR with subjective timescales in Figure 5.

### C.3 VPR ON THE BOUNCING BALLS DATASET

To further illustrate the effectiveness of the VPR's event detection system, we additionally train it using a more dynamically complex dataset, analogous to the one used in Kim et al. (2019). The results demonstrate that VPR outperforms the VTA model (Kim et al., 2019) in determining event boundaries, as seen from Figure 13. The resultant representational properties of VPR with respect to the two temporal factors of variation (position and colours) proved to exhibit the same hierarchical disentanglement features discussed in Section 4.2 and demonstrated using samples taken from the different latent levels of the model in Figure 14.

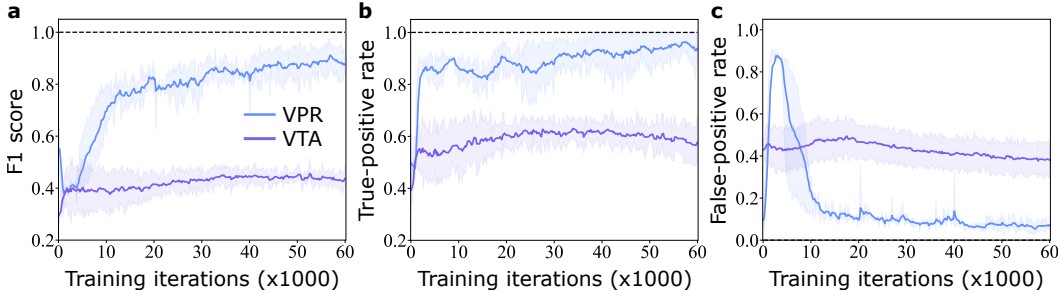

Figure 13: Comparison of VPR against VTA on the task of event detection using the Bouncing Balls dataset. (**a**) F1 score. (**b**) True-positive rate. (**c**) False-positive rate. Shaded region indicates one standard deviation using 5 different seeds.

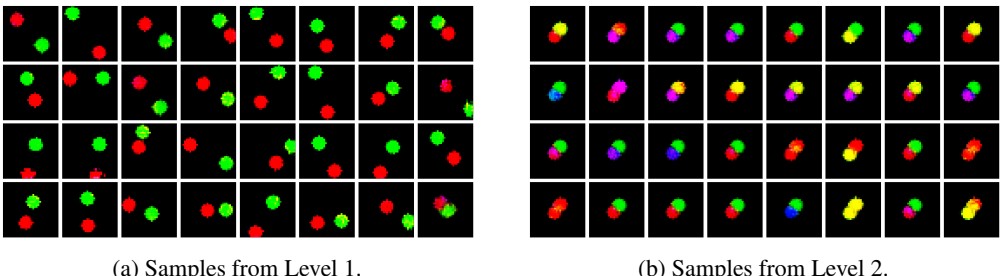

(a) Samples from Level 1.        (b) Samples from Level 2.

Figure 14: Random samples taken from the different levels of VPR. Similar to the analysis in Section 4.2, it is evident that VPR learns to represent the position of the balls in Level 1 and their individual colours in Level 2 of the hierarchy.

## C.4 CU SENSITIVITY ANALYSIS

As shown, the CU threshold is an important component of the event detection mechanism that incorporates two main hyperparameters – window size of the moving average $\tau_w$ and threshold weight $\gamma$. Using the Moving Ball dataset, we evaluate the sensitivity of the VPR's performance to the different values of these two parameters in Figure 15. The graphs indicate an average achieved F1 score by VPR using the specified parameter value. VPR's performance for each of the parameter values is averaged across all VPR instances trained using all of the values of the other parameter. In total, 100 different models were trained (5 sliding window sizes × 4 threshold weights × 5 runs each).

We find that changing the size of the moving average window does not have a significant effect on the model's performance. This is not the case for the threshold weight parameter that resulted in the deterioration of VPR's performance when $\gamma$ was increased to $1.2$. This is likely due to the fact that a significant increase in the value of $\gamma$ results in a more sparse event detection that poorly matches the rate of feature changes. Overall, however, we find that the two hyperparameters are quite robust, as the same set of values ($\gamma = 1.1$ and $\tau_w = 100$) was used across all of the datasets presented in the paper.

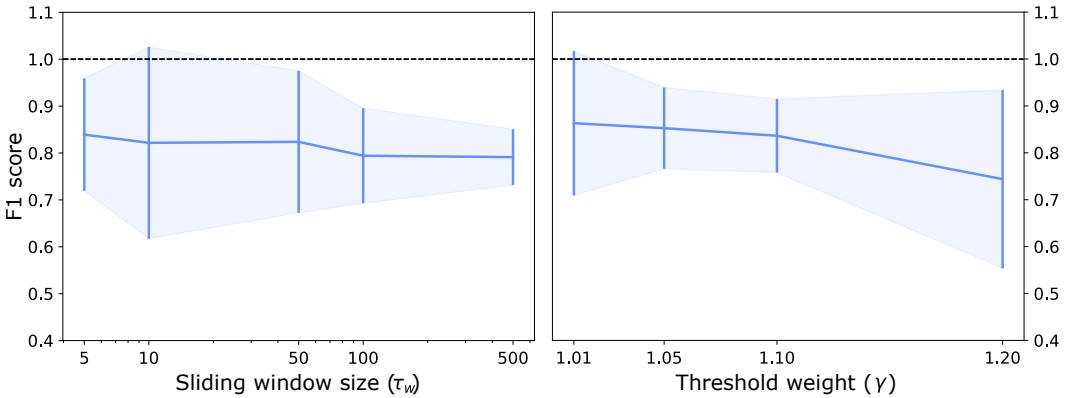

Figure 15: Average achieved F1 score against the two hyperparameters of the CU threshold: window size $\tau_w$ and threshold weight $\gamma$. Shaded region indicates one standard deviation.

C.5 Decomposing the KL terms of the *CE* criterion

The KL terms of the *CE* criterion can be conveniently decomposed into the entropy and the cross entropy terms,

$$D_{KL}(q_{ch}||p_{ch}) = -\underbrace{H(q_{ch})}_{\text{entropy}} + \underbrace{\mathbb{E}_{q_{ch}}(\log p_{ch})}_{\text{cross entropy}}, \tag{9}$$

$$D_{KL}(q_{st}||p_{st}) = -\underbrace{H(q_{st})}_{\text{entropy}} + \underbrace{\mathbb{E}_{q_{st}}(\log p_{st})}_{\text{cross entropy}}, \tag{10}$$

where *CE* inequality can be re-written as,

$$-H(q_{st}) + \mathbb{E}_{q_{st}}(\log p_{st}) > -H(q_{ch}) + \mathbb{E}_{q_{ch}}(\log p_{ch}). \tag{11}$$

By recording the differences between the entropy and cross entropy components (see Fig. 16), we find that decision-making is largely dominated by the cross entropy terms. This implies that the 'decision-making' inequality above can be approximated as,

$$\mathbb{E}_{q_{st}}(\log p_{st}) > \mathbb{E}_{q_{ch}}(\log p_{ch}), \tag{12}$$

since $H(q_{st}) - H(q_{ch}) \approx 0$. This interprets the model's CE event detection as a selection of a more well-predicted state based on the *static* or *change* assumptions.

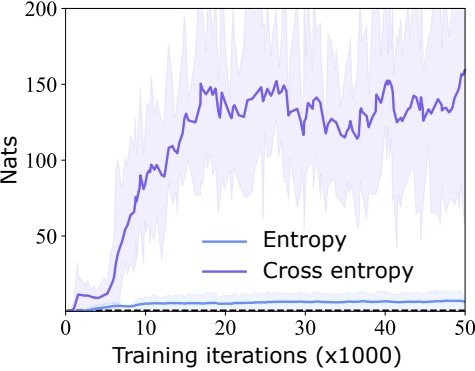

Figure 16: Comparison of the difference in entropies $|H(q_{ch}) - H(q_{st})|$ and cross entropies $|\mathbb{E}_{q_{ch}}(\log p_{ch}) - \mathbb{E}_{q_{st}}(\log p_{st})|$ of the two competing hypotheses denoting whether or not a state change has occurred at timestep $t$ and layer $n$. Data for this graph was acquired by training multiple instances of VPR with the Moving Ball dataset and for level $n = 2$.

C.6 Event detection in the Miniworld Maze.

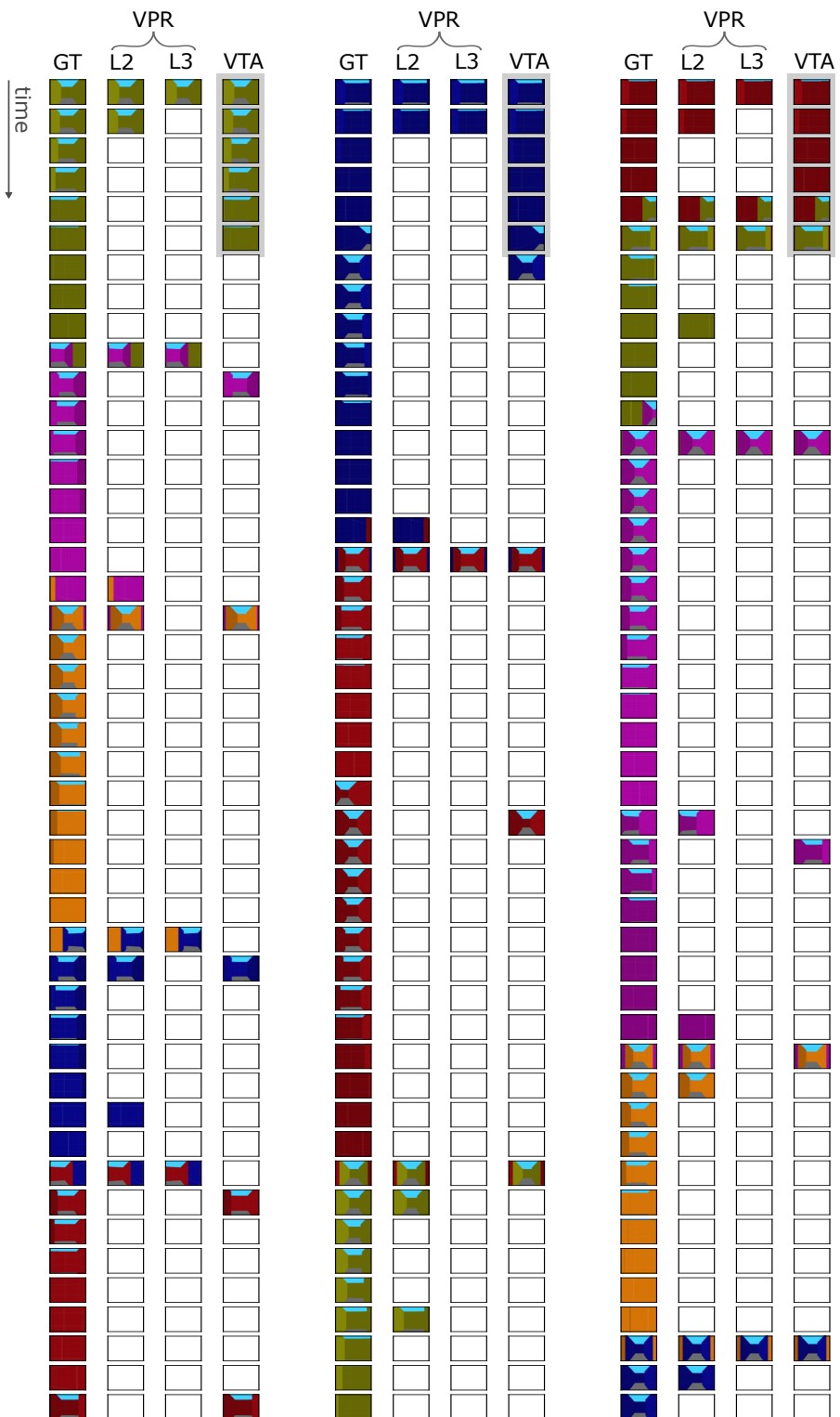

Figure 17: Comparison between VPR and VTA event detection using random sequences from the MiniWorld Maze dataset. GT denotes the ground-truth sequence. L2 and L3 denote events that have been captured by the second and third level of the VPR model respectively. L2 is able to detect events that correspond to colour changes, L3 detects more sparse events relating to the agent's location in the maze, while the VTA model detects turns between corridors. Best seen in the electronic version of the manuscript.

