# OpenReview forum: "Variational Predictive Routing with Nested Subjective Timescales"
_ICLR.cc/2022/Conference — ICLR 2022 Poster_

### Official Review · Reviewer_A5x3 · 2021-10-27

**Correctness:** 4
**Technical Novelty And Significance:** 3
**Empirical Novelty And Significance:** 3
**Recommendation:** 8
**Confidence:** 3

**Main Review:**

strengths
---------
VPR represents a significant advance over previous models like CW-VAE (which requires a fixed "event detection" interval) and VTA (where event detection must be learned separately).

For the most part the method is clearly explained; Figs 1 and 2 are very helpful.

The 1D synthetic dataset is useful for understanding the method in a simple setting. Do similar results to Fig. 4 emerge from the other datasets as well?

The layerwise rollouts are an interesting way to probe the representations at each layer.


concerns
--------
I still don't fully understand the event prediction (specifically CE). Though the paper does say in words what D_{st} and D_{ch} are (which is much appreciated), the fact that their comparison is a reasonable way to define a predictable event is not clear to me. Does D_{ch} represent a difference we would expect from the structure of the model with/without seeing new data? Why does it make to compare D_{st} to this? Spelling this out a bit more clearly would help explain this crucial piece of the method.

How is the F1 score computed for change detection? Do you arbitrarily define successive events (in both the data and the model) as 0->1->0->1->... and then compute F1 on these binary sequences? Would be useful to spell this out more clearly.

The current figure layout is convenient space-wise, but makes following the results harder than it needs to be.

Fig 4 legend is hard to parse.
* "CE" appears to be "CE with CU enabled"; what then is CE+CU?
* "CU disabled" appears to be "CE with CU disabled"; what then is CU? CU only, but in the context of "CE with CU enabled"?
* why is there no CU curve in Fig 4A?
* why do 4A and 4C show different models? would make more sense to show the same models under the noise and no-noise condition

Fig 5A - do you have an intuition for why the color changes in L2 on every frame? Given the frequency of these events (on wall bounce or otherwise with probability 0.1) I would have expected the model to learn slower transitions between colors.

Experiments are run with other baselines, but inconsistently. Running VTA on the 1D synthetic data could be instructive, and would especially be interesting to see how it performs under different noise levels. It should also be easy enough to add a CW-VAE curve to Fig. 3B. This will perform terribly, but that also opens up an opportunity to revisit the fundamental differences between VPR and CW-VAE and why the new model is an advance. I think both rollouts and random samples should be added in appendix figures for the CW-VAE and VTA. In general, the results section could start off with an explanation of the datasets (as it does already) AND an explanation of the baseline models, and why those models were chosen. That gives a bit more structure to (and motivation for) the comparisons made later on.

How will this model behave when there are slow, continuous changes in high-level features, rather than discrete changes? For example, what if colors slowly cycled through hue space in the Moving Ball dataset (much more slowly than position)? Would kind of features would VPR learn at L2? Not suggesting this as an experiment to perform, but might be an interesting direction to address in the discussion.


**Summary Of The Paper:**

The paper introduces VPR, an unsupervised hierarchical latent variable model. The latent representation is organized into a temporal hierarchy, with deeper layers representing features with slower rates of change. This hierarchy is enabled by an event detection mechanism that finds expected events in the input using VPR's internal transition model, and finds unexpected events through changes in the latent distributions at each level of the hierarchy. The paper demonstrates VPR's performance on unsupervised event boundary discovery, and how it is able to learn disentangled representations, on a variety of synthetic video datasets.

**Summary Of The Review:**

This paper presents a novel, well-motivated method for inferring temporal hierarchy in latent representations of sequence data. I lean towards accepting this paper, and would be willing to further increase my score if some of the concerns above (especially regarding the experiments) are addressed in revision.

---

> ### Author Response · Authors · 2021-11-23
> **Response to Reviewer A5x3 (Part 1)**
>
> Dear Reviewer,
>
> Thank you for your review!
>
> It is great to hear that you think our model is a significant advance over fixed-interval models or parametrised event detection models. Certainly, we also believe that flexible (or subjective) and nested timescales are a very promising direction of research in hierarchical generative modelling, and non-parametric detection mechanisms can very much help with that, as we demonstrated in the paper. We hope that the following comments can answer your questions.
>
> > "**The 1D synthetic dataset is useful for understanding the method in a simple setting. Do similar results to Fig. 4 emerge from the other datasets as well?**"
>
> Yes! In Figure 3b, we demonstrate a very similar behaviour on the Moving Ball dataset (mentioned on Page 7). In particular:
> - When only CE is used, the F1 score of the model is slowly rising to a high value (see also Fig.4c);
> - When only CU is used the F1 score stays constant at a lower value for most of the training duration;
> - When both CE and CU are enabled, we achieve the optimal performance (see also Fig.4a).
>
>
> > "**I still don't fully understand the event prediction (specifically CE). Though the paper does say in words what D_{st} and $D_{ch}$ are (which is much appreciated), the fact that their comparison is a reasonable way to define a predictable event is not clear to me. Does $D_{ch}$ represent a difference we would expect from the structure of the model with/without seeing new data? Why does it make to compare $D_{st}$ to this? Spelling this out a bit more clearly would help explain this crucial piece of the method.**"
>
> Thank you for pointing this out and for the suggestion! The main reason why the CE mechanism works well is because taking the *minimum* of the two terms, $D_{st}$ and $D_{ch}$, is approximately equivalent to selecting the hypothesis that maximizes the likelihood of the latent state given a new observation. An event is thus detected when the *change* hypothesis (assumption that a change in the features has occured) results in the maximum likelihood.
>
> We understand that this intuition behind the CE criterion may not have been explained clearly enough in the first version of the manuscript. For this reason, we have now amended the discussion section titled "Expected and unexpected events", as well as added a new supplementary section (C.3 - Decomposing the KL terms of the CE criterion) discussing this point in more detail.
>
>
> > "**How is the F1 score computed for change detection? Do you arbitrarily define successive events (in both the data and the model) as 0->1->0->1->... and then compute F1 on these binary sequences? Would be useful to spell this out more clearly.**"
>
> Each dataset contains temporal factors of variation that change over time. When such changes occur (e.g. ball changes its colour), we do indeed record it in a binary sequence, such that "1" indicates a colour change and "0" no change in the factor.
>
> At the same time, during training, the model employs the event detection mechanism to decide on whether a hierarchical state should be updated. When this happens, the model's decision is similarly recorded in a binary sequence using the same categorisation as mentioned above. After that, the ground-truth and model's event sequences are compared using the metric of F1 score.
>
> > **"The current figure layout is convenient space-wise, but makes following the results harder than it needs to be."**
>
> Unfortunately, we have tried our best to choose a different layout; however, it seems impossible given the space constraints. We do apologise if it makes it less accessible.
>
> > **"Fig 4 legend is hard to parse."**
>
> We apologise for the confusion caused by this admittedly complex legend.
>
> Figure 4 examines the performance of two different models -- the default VPR (with both CE and CU criteria at work) and a version of VPR where event detection is realised using only the CE criterion. The performance of the default VPR model is shown with the black curve of Fig.4a, while the performance of the VPR model without CU is shown with the pink curve of the same figure. Additionally, we further break down the performance of the default VPR model to assess the extent to which each of the criteria -- CE in blue and CU in orange -- contributes to the overall performance of the model (in black). Importantly, these two curves are made by training the default VPR model but recording only events detected by each of the criteria separately. Finally, we have now also added a comparison with VTA in Fig.4d.
>
> The legend of this figure is now changed to emphasize this distinction more. Thank you for pointing this out to us.

---

> > ### Author Response · Authors · 2021-11-23
> > **Response to Reviewer A5x3 (Part 2)**
> >
> > > **"Fig 5A - do you have an intuition for why the color changes in L2 on every frame? Given the frequency of these events (on wall bounce or otherwise with probability 0.1) I would have expected the model to learn slower transitions between colors."**
> >
> > Thank you for this question -- it is an important one. Indeed, this rollout (along with the rollouts of L2 and L3 in the 3D Shapes dataset) illustrate one of the most important aspects of the model. As explained, VPR is capable of detecting events using the CE and CU criteria. What this implies, however, is that it also learns to transition between these events, since every detected event causes the model to update its relevant hierarchical state. For example, in the Moving Ball dataset, VPR learns to represent the colour of the ball in L2 by means of the event detection system. This means that the latent state in L2 will update (or transition) every time a colour change is detected. By consequence, the transition model will learn this transitioning, which is what can be observed property. We refer to this emergent behaviour of the model as the ability to produce *feature-specific* and *jumpy* rollouts.
> >
> >
> >
> > > **"Experiments are run with other baselines, but inconsistently. Running VTA on the 1D synthetic data could be instructive, and would especially be interesting to see how it performs under different noise levels."**
> >
> > Thank you for pointing this out and we completely agree. For this reason, we have now run further experiments with the selected baselines to make the presentation of our results more consistent (see our general response for the changes made to the manuscript).
> >
> > Regarding VTA on the 1D synthetic data, we performed the recommended analysis by training 30 instances of the VTA model (5 per noise level), and we have now included a comparsion of its performance under the different noise levels in Fig.4d.
> >
> >
> >
> > > **"It should also be easy enough to add a CW-VAE curve to Fig. 3B. This will perform terribly, but that also opens up an opportunity to revisit the fundamental differences between VPR and CW-VAE and why the new model is an advance."**
> >
> > We have now included a comparison between VPR and a fixed-rate (clockwork) version of VPR, the fixed-rate structure of which is analogous to CW-VAE. We termed it CW-VPR -- see Fig.3c-d. Thank you for the suggestion, as we also found it a good idea to illustrate the key distinction between fixed-rate models and our proposed approach.
> >
> >
> >
> > > **"In general, the results section could start off with an explanation of the datasets (as it does already) AND an explanation of the baseline models, and why those models were chosen. That gives a bit more structure to (and motivation for) the comparisons made later on."**
> >
> > Great idea. We have now included a paragraph introducing the baseline models after the explanation of the datasets. Thank you for the recommendation.
> >
> >
> >
> > > **"How will this model behave when there are slow, continuous changes in high-level features, rather than discrete changes? For example, what if colors slowly cycled through hue space in the Moving Ball dataset (much more slowly than position)? What kind of features would VPR learn at L2? Not suggesting this as an experiment to perform, but might be an interesting direction to address in the discussion."**
> >
> > This is also a very good question. The fact that changes in features occur instantaneously, rather than gradually, as in the case of hues in the Moving Ball dataset, is indeed a crucial factor for creating the hierarchical structure of VPR.
> >
> > If the ball changed hues gradually, with a new tertiary colour being introduced in each video frame, the current version of VPR would classify this as a series of changes and would learn to represent the ball colour in L1, even if the full cycle was much slower than ball position.
> >
> > In future versions of VPR, we plan to experiment with (a) discrete latent spaces and (b) classification mechansims. This would allow VPR to create distinct categories of features that change continuously and thus to alleviate this effect.
> >
> >
> >
> >
> > Thank you once again for your review -- we are looking forward to hearing back from you!

---

> > > ### Author Response · Authors · 2021-11-28
> > > **To Reviewer A5x3**
> > >
> > > Dear Reviewer A5x3,
> > >
> > > Thank you once again for your review of our work. As the discussion period is approaching its end, we would be grateful if you could confirm whether our responses and the additions we have made to the manuscript addressed your concerns, and let us know if any issues remain.

---

> > > > ### Comment · Reviewer_A5x3 · 2021-11-30
> > > > **response to authors**
> > > >
> > > > I appreciate the updates the authors have made to the manuscript, particularly the additional baselines. I think these updates have further clarified the method itself, as well as its performance gains over previous methods. I am increasing my score from 6 to 8.

---

> > > > > ### Author Response · Authors · 2021-11-30
> > > > > **To Reviewer A5x3**
> > > > >
> > > > > Dear Reviewer A5x3,
> > > > >
> > > > > We are very pleased to hear that -- thank you for raising your score!

---

### Official Review · Reviewer_ovNG · 2021-11-03

**Correctness:** 3
**Technical Novelty And Significance:** 4
**Empirical Novelty And Significance:** Not applicable
**Recommendation:** 6
**Confidence:** 4

**Main Review:**

##########################################################################

Pros:

-I think that exchanging ideas between neuroscience and machine learning is a great way to develop novel generative models with new capabilities such as VPR. I felt as if this work was also indirectly motivated by a desire to improve our knowledge about how to design better world models, which is a really important topic for model-based reinforcement learning.  This paper has the potential to be of significant interest to that community as well.

-I was surprised at how cleanly VPR is able to learn the dynamics of distinct latent factors in each level, as evidenced clearly by the qualitative results. Despite the fact that the datasets are by and large toy-like and synthetic, this was still great to see.

-The analytical experiments are well-thought-out and validate the role that the various components of the VPR play (e.g., in Figures 3,4, and 7).

-The presentation is solid---I provide some suggestions to improve it below. The visualized qualitative results are really helpful for quickly grasping how VPR works and what it is capable of learning.

##########################################################################

Cons:

-The introduction motivates this work as one which can be “beneficial for predicting future sequences”, and the beginning of the experiments section even lists “future event prediction” as a task to be studied. However, I was disappointed to find this is missing from the model design (as explained in Section 5: Event duration estimation) and is missing from the experiments.

-I found it strange that TD-VAE  by Gregor et al. (which is cited in the introduction) is not discussed again in more detail under “Hierarchical generative models” in Section 2. This model learns a jumpy state-space model, supports a hierarchy of latent variables with transition functions operating at different time scales, and can handle future prediction. On the surface level, it does not have the explicit event detection mechanism of VPR, but it still seems capable of learning a transition function over “important states”.

-I would have liked to see at least one comparison between TD-VAE and CW-VAE and VPR (if/when able to support future prediction) to judge VPR’s ability to be useful as a world model and perhaps to highlight where it excels whereas these previous methods do not. The experiment comparing fixed-interval versions of VPR in terms of feature disentanglement (Figure 7) is illuminating but is not really satisfying for this since it is not evaluating on long-term prediction. Without such an experiment and a comparison against these rigorous baselines, the maturity of the work still feels to be at somewhat of an exploratory stage.

-I would have liked to see more analysis of the Miniworld Maze results as it is the most visually challenging environment and the dynamics of the changing latent factors are less hand-engineered. Comparing VPR against VTA here would be helpful.

-Section 3 needs some work to make the presentation more clear:

--I couldn’t understand Figure 2. Figure 8 in the appendix was much more clear.

--I didn’t see where $p(o_t | s_t^{1:N})$ is defined.

--The usage of $t$ and $\tau$ got very confusing.  It would be good to stick with one throughout the paper, if possible.

--I believe the posterior models should be conditioned on $s_{\tau -1}^{>n}$, not $s_{\tau}^{>n}$.

-A discussion on how $\gamma$ and $\tau_w$ are selected, and how sensitive the model is to these variables, is missing. These values should be affected by the size of the latent dimension, which in turn affects the value of the KL used for event detection.

##########################################################################

Some suggestions for improvement:

-In Figure 3, use a), b), c), and d) to describe sub-figures

-In figure captions where the figures have error bars or confidence intervals, tell how many seeds/runs are used in the caption

**Summary Of The Paper:**

This paper introduces Variational Predictive Routing (VPR), a spatiotemporal deep generative model for hierarchical video representation learning. It is motivated by a desire to implement predictive routing---a model of cognition---within a deep generative modeling framework. VPR automatically detects expected and unexpected changes at each level of the latent hierarchy, which it uses to “route” bottom-up and top-down signals. Interactions between bottom-up (from observations) and top-down signals (from the prior) are crucial for implementing this hierarchy and are a core aspect of the predictive routing framework. Each level of the hierarchy learns a time-agnostic transition model for a distinct dynamic latent factor. By design, “slower” dynamic features are modeled at higher levels of the hierarchy and “faster” dynamic features are modeled at lower levels. Experiments on synthetic video datasets validate the success of VPR at implementing predictive routing within a variational representation learning framework.

**Summary Of The Review:**

I currently believe this paper is borderline and I am leaning to recommend to reject.
This is because I think the way the paper frames VPR as being comparable with prior methods like CW-VAE and as a potentially-useful framework for long-term prediction is promising, yet more work is needed (described in my review) to take this work from an exploratory stage to ready for publication. I look forward to hearing from the authors during the rebuttal period on this matter.
==========================================================================
Update after author response: The authors have provided satisfactory responses to my concerns and revised the paper with extra results and discussions accordingly. I have increased my score to a 6.

---

> ### Author Response · Authors · 2021-11-23
> **Response to Reviewer ovNG (Part 1)**
>
> Dear Reviewer,
>
> Thank you for your review!
>
> We are very happy to hear you are also of the opinion that neuroscience can serve as inspiration for building novel generative architectures such as ours. We also appreciate your recognition of our future research direction, by which this paper was partially driven; indeed, building better world models for model-based reinforcement learning is something we will be looking at with VPR in the future. Similarly, we are encouraged to learn you were pleased with the VPR's disentanglement capabilities and approve of our analytical experiments. In what follows, we hope to address some of the questions you raised and would be happy to hear what you think.
>
>
>
> #### Main comments
>
> > "**I was disappointed to find [future event prediction] is missing from the model design (as explained in Section 5: Event duration estimation) and is missing from the experiments.**".
>
> Thank you for this comment! It is a very important point of clarification we wish to make. Section **5** talks about performing rollouts over the physical timescale of the environment, which, unfortunately you are right, our model does not support at the moment. However, there is a noteworthy difference between physical-timescale rollouts and future event prediction. The latter refers to the model's ability to perform accurate layerwise rollouts with respect to the learned factors of variation of the dataset, since the changes in those factors of variation is what constitutes an *event* (e.g. colour change of a ball).
>
> This distinction is important because we do demonstrate VPR's ability for accurate event prediction. Specifically, layerwise rollouts in Figure 5 show VPR's ability to predict future events -- a ball colour change in the *Moving Ball* and wall/object colour changes in the *3D Shapes*. Similarly, we reported the accuracy of event prediction for 3D Shapes (Section **4.2** -- paragraph 2). Nevertheless, we understand that this may not have been elucidated well enough and certainly not emphasised; therefore, we decided to add a small section (4.4) to clarify this point more thoroughly. There, we additionally report event prediction accuracy for the Moving Ball dataset, as well as make an indicative comparison with a video prediction model CW-VAE [Saxena et al., 2021]. We hope this addresses your concern.
>
>
>
> > "**I found it strange that TD-VAE by Gregor et al. (which is cited in the introduction) is not discussed again in more detail [...]. This model learns a jumpy state-space model, supports a hierarchy of latent variables with transition functions operating at different time scales, and can handle future prediction. On the surface level, it does not have the explicit event detection mechanism of VPR, but it still seems capable of learning a transition function over “important states”.**"
>
> Thank you for this comment -- we would be happy to clarify this. Naturally, we have looked at TD-VAE as a model that could be reviewed (and even compared to) in more detail. However, we found a couple of important differences that led us to choose other models, instead. (Needless to say, this is also largely due to the limited space we had.)
>
> Specifically, we wanted to focus our review on two key properties of a hierarchical model -- event detection (or discovery of temporal structure) and nested timescales. With respect to the former, we found that TD-VAE does not have any explicit mechanism for event boundary detection (as you mention). Instead, it learns to transition between states randomly sampled from a uniform distribution (Sections 4.1 and 5). Although the authors demonstrate the nice properties of jumpy predictions that are useful in reinforcement learning, we wanted to focus specifically on the area of representation learning and temporal structure discovery. Therefore, we felt that papers like HM-RNN [Chung et al., 2017] or VTA [Kim et al., 2019] were more in line with this direction. With respect to nested timescales, we also found that although TD-VAE does support a hierarchy of latent variables, it has no meaningful way to enforce a hierarchy of nested timescales. Perhaps, this is a misunderstanding on our side, but we couldn't find where the authors talk about different levels of TD-VAE operating at different timescales, which is why we decided to place more focus on a model like CW-VAE [Saxena et al., 2021] in our review.
>
> Still, we think TD-VAE is a great approach that would make a great world model comparison in the future work. We are currently working on applying VPR in reinforcement learning to see how it could be beneficial for agent planning, for which we believe TD-VAE will make a great comparison. We hope this clarifies our thinking on this subject!

---

> > ### Author Response · Authors · 2021-11-23
> > **Response to Reviewer ovNG (Part 2)**
> >
> > > "**I would have liked to see at least one comparison between TD-VAE and CW-VAE and VPR (if/when able to support future prediction) to judge VPR’s ability to be useful as a world model and perhaps to highlight where it excels whereas these previous methods do not.**"
> >
> > We agree that this would be an interesting avenue for future work! As mentioned, this paper was partly motivated by the desire to create a novel world model architecture. However, it is important to clarify that this was not the focus of the current paper; rather, it partly served as inspiration. As you rightfully pointed out, there are no explicit experiments to support the claim that VPR is a useful *world model* (perhaps, only hints at this point). Instead, the main theme of the paper and its main contributions relate to the emergent representational properties of VPR (e.g. disentangling temporal factors of variation) and unsupervised event discovery.
> >
> > Nevertheless, here we would also like to refer to our first comment, explaining what we mean by the distinction between the physical-timescale rollouts and future event prediction. As discussed in Section 5, our model does not support physical timescale rollouts, but our analysis illustrates VPR's ability to do the latter. To make this point more distinct, we have added an extra section (4.4) dedicated solely to future event prediction, including a comparison with CW-VAE.
> >
> >
> >
> > > "**I would have liked to see more analysis of the Miniworld Maze results**"
> >
> > Great suggestion! We have added an interesting comparison of event detection in the MiniWorld Maze between VPR and VTA -- see Appendix **C.6**. In short, we provide an illustrative example of what events are being detected across the VPR's hierarchy.
> >
> >
> > > "**A discussion on how and are selected, and how sensitive the model is to these variables, is missing**"
> >
> > We absolutely agree with you here. We have now added a sensitivity analysis, which you can find in Appendix **C.4**.
> >
> >
> > > "**This is because I think the way the paper frames VPR as being comparable with prior methods like CW-VAE and as a potentially-useful framework for long-term prediction is promising, yet more work is needed (described in my review) to take this work from an exploratory stage to ready for publication**"
> >
> >
> > We hope that the above clarifications about the paper's main theme have alleviated some of your concerns on this matter. While we do see physical-timescale prediction as a great avenue for future work, in this paper we wanted to focus on representation learning and unsupervised event discovery. Nevertheless, as mentioned, future *event* prediction is part of the analysis as a naturally emerging property of the hierarchical transition models operating over different timescales and temporal features. It can also be viewed, and is in fact framed in the paper, as the ability to perform feature-specific and jumpy rollouts. We hope that the extra section (4.4) and the analysis have helped in clarifying this point. We are looking forward to hearing back from you on this subject.
> >
> >
> >
> > #### Minor comments
> > - Figure 2 illustrates the interactions between the computational blocks, specifically their means of 'communication' through the deterministic variables (shown in more detail in Figure 1). If you don't mind, it would be great if you could point us to the things you found confusing, as we are not sure how to make the figure more accessible and given that other reviewers seemed to have no issue with it. Thank you!
> > - $p(o_t|s_t^{1:N})$ is represented by Eq. 7 $o_{\tau} = f_{rec}(c_\tau^0)$ -- the reconstruction model. Recall that similar to a recurrent network that accumulates information about the past states using a deterministic hidden state, $d$, top-down generation accumulates information about hierarhical states by means of variable $c$. Therefore, $c^{0}_\tau$ represents all of the hierarchical states at timestep $\tau$ -- $s^{1:N}_\tau$.
> > - We have changed the time subscripts to $\tau$ to remove any confusion. Thank you for the suggestion!
> > - The posterior models are conditioned on $s_\tau^{>n}$ (as opposed to $s^{>n}_{\tau-1}$) since the higher levels posteriors are inferred first and are used in the inference of lower-level states.
> > - We have added a sensitivity analysis on the parameters $\gamma$ and $\tau_w$ of the CU criterion in Appendix **C.4**. Thank you!
> >
> >
> > Once again, we appreciate such a detailed review and are looking forward to hearing back from you! Let us know if we were able to address your concerns.

---

> > > ### Author Response · Authors · 2021-11-28
> > > **To Reviewer ovNG**
> > >
> > > Dear Reviewer ovNG,
> > >
> > > Thank you once again for your review of our work. As the discussion period is approaching its end, we would be grateful if you could confirm whether our responses and the additions we have made to the manuscript addressed your concerns, and let us know if any issues remain.

---

> > > ### Comment · Reviewer_ovNG · 2021-11-29
> > > **Response to authors**
> > >
> > > Thank you for updating the paper with additional discussions and results. The responses are by and large satisfactory and I have increased my score accordingly.
> > >
> > > Thank you for clarifying the distinction between future event prediction and physical-timescale rollouts. Section 4.4 and the revised section 5 now make this much more clear. Also, thanks for your explanation about TD-VAE. Still, I would recommend adding somewhere in the paper that a promising next step for this line of work is to explore using VPR as a world model for RL where it can be compared with relevant world models like TD-VAE. This would help to position this work better within the relevant literature, I think.
> > >
> > > Thanks for adding the results in Appendix C.6 for Miniworld Maze. It looks like L2 and L3 occasionally fire on the same events and L3 occasionally misses some events, which I imagine correspond to false positives and false negatives. This helps a lot. Thanks for adding the sensitivity analysis results in Appendix C.4 as well.

---

> > > > ### Author Response · Authors · 2021-11-29
> > > > **To Reviewer ovNG**
> > > >
> > > > Dear Reviewer ovNG,
> > > >
> > > > We are happy to hear this -- thank you for increasing your rating!
> > > >
> > > > With respect to TD-VAE, if the paper is accepted, we will certainly do our best to accomodate a short discussion on the future line of work in the camera-ready version of the manuscript.

---

### Official Review · Reviewer_RjVA · 2021-11-03

**Correctness:** 4
**Technical Novelty And Significance:** 3
**Empirical Novelty And Significance:** 3
**Recommendation:** 6
**Confidence:** 3

**Main Review:**

I like the general idea of modeling change detection as the difference in divergences, I think this indirection is novel.
I miss some more details on the following things:
* What is the generative model for the change detection? To my understanding, change detection is modeled as an interplay between the inference  model and the generative model, but shouldn't there be some part in the generative model alone that is handling 'changes'?
* Can you expand on the "non markovian property' of the model (paragraph before eq 5)? I assume this makes s_t independent from all x_t', t' > t?
* A pseudo algorithm would be very helpful to see exactly where/how the decision to stop propagating information upwards happens.
* It would be great (but admittingly hard) to have more expressive datasets.
* Related to datasets: In a real world setting, I believe that one can have maybe a bit of time to look at a prefix part of a sequence to be handled, and needs to 'extract' the core properties of that sequence with respect to event onsets. How could that be handled? I assume this becomes some sort of meta learning?
* I think you should cite "Neural Sequnce Chunker" by J. Schmidhuber and Unsupervised Real-Time Control through Variational Empowerment  by M. Karl et al.

**Summary Of The Paper:**

Variational Predictive Routing (VPR) models continuous data as a hierarchical renewal process. More specficially, it is a hierarchical generative model that organizes the temporal hierarchy based on the rates of change of its latent representations. The core contribution hereby is that event detection is formulated as a comparison between semantically consistent KL divergences (or, in the case of CU, a moving average of KL divergences). VPR is time agnostic and can model a wide variety of temporal structures. Compared to a baseline method (VTA), VPR detects changes in synthetic datasets with high accuracy.

**Summary Of The Review:**

I believe the paper has a very nice core idea, and a well thought out 'harness' around that idea. I miss more conceptualized explanations, and would have loved more experiments (and also more comparisons to comparable models).

---

> ### Author Response · Authors · 2021-11-23
> **Response to Reviewer RjVA (Part 1)**
>
> Dear Reviewer RjVA,
>
> Thank you for your review of our paper! We are very happy that you share our enthusiasm about the main idea of the paper.
>
>
> ### Main comments
>
> > "**What is the generative model for the change detection? To my understanding, change detection is modeled as an interplay between the inference model and the generative model, but shouldn't there be some part in the generative model alone that is handling 'changes'?**"
>
> You are absolutely correct in saying that change detection is realised only by operating on the beliefs produced by the generative and inference models. But importantly, VPR does not possess any additional components that handle changes (we like to think that this is one of the beautiful aspects of our approach).
>
> Instead, changes (or events) are solely inferred by computing the divergence quantities (specified in Criteria U and E) using the *existing* parts of the VPR's model -- i.e. only using the posterior, prior, and transition models (Eqs. 2-4). The only things that change when employing these models for computing the divergences are the *inputs*. For instance, when calculating posteriors $q_{st}=q_\phi(s^n_{\tau+1} | ...)$ and $q_{ch} = q_\phi(s^n_{\tau+1} | ...)$, the same posterior model $q_\phi$ is used -- it is the difference in the inputs that makes a difference.
>
> To emphasise this point more, we have added parameter subscripts to indicate that the same model components are employed for event detection. We hope we could clarify this for you!
>
>
>
>
> > "**Can you expand on the "non markovian property' of the model (paragraph before eq 5)? I assume this makes s_t independent from all x_t', t' > t?**"
>
> Certainly! By non-markovian property we wanted to highlight the dependencies of the random variables in the factorised form of the generative model. Specifically, in the temporal direction, we specify a distribution over a latent state $s^n_\tau$ as being conditioned on the previously inferred states of the same level $s^n_{<\tau}$. This is common in the literature when using a recurrent transition model, and is being realised by the hidden state of the model ($d$ in our case). In the top-down direction, we also wanted to emphasise the conditioning of a latent state $s^n_\tau$ on all of the inferred states above (at the same timestep), $s^{>n}_\tau$. This is similarly the case for the reconstruction term, $p(o_t|s^{1:N}_t)$. We believe this is an interesting component of the model, as it allows for a richer representational power. In particular, it allows the model to combine the information stored in the different hierarchical states both to produce a reconstruction image via $p(o_t|s_t^{1:N})$, and to produce lower-level states during generation, $p(s_t^n|s^{>n}_t)$. This idea is similarly explored in a powerful image model NVAE [Vahdat and Kautz, 2020].
>
>
>
>
> > "**A pseudo algorithm would be very helpful to see exactly where/how the decision to stop propagating information upwards happens.**"
>
> We have added a pseudocode section in Appendix **B.1** that concerns the process of bottom-up  (event detection) and top-down (inference) pathways. We hope this can be of help for understanding the mechanism.
>
>
>
>
> > "**It would be great (but admittingly hard) to have more expressive datasets.**"
>
> This is a great suggestion and indeed a great avenue for future research using VPR. In the current manuscript we tried to select interpretable datasets that could be used for a straight-forward comparison and a careful examination of the basic properties of the learned latent representations -- such as spatiotemporal disentanglement and factor-specific transitioning. Nevertheless, we have added another, more dynamically complex dataset -- Bouncing Balls (Appendix **C.3**). Although it may not necessarily be considered more 'expressive', we wanted to demonstrate that VPR similarly works on a more challenging dataset (which is also used in the VTA paper).

---

> > ### Author Response · Authors · 2021-11-23
> > **Response to Reviewer RjVA (Part 2)**
> >
> > > "**Related to datasets: In a real world setting, I believe that one can have maybe a bit of time to look at a prefix part of a sequence to be handled, and needs to 'extract' the core properties of that sequence with respect to event onsets. How could that be handled? I assume this becomes some sort of meta learning?**"
> >
> > Interesting idea! We think you are correct in saying that something like this would likely relate to meta learning. It seems that this would require a separate parametrised model for extracting such patterns from the provided context. Perhaps, this could actually be done in a form of additional priors, which could be integrated into the definition of a generative model and computed based on the extracted patterns one typically sees in the considered dataset (e.g. distance between events). If viewed from this perspective, it is possible to integrate it into the existing detection mechanism of VPR, since it relies on the computation of the relevant belief states, semantically distinguished by the *change* and *static* assumptions. Conditioning the computation of these states on the extracted priors could be one of the ways for doing so. Thank you for such a curious idea!
> >
> >
> >
> > > "**I think you should cite "Neural Sequnce Chunker" by J. Schmidhuber and Unsupervised Real-Time Control through Variational Empowerment by M. Karl et al.**"
> >
> > Thank you for the suggestion! We now cite Neural Sequence Chunker in the introduction. With respect to Unsupervised Real-Time Control through Variational Empowerment by M. Karl et al., we were not entirely sure -- would it be possible for you to briefly explain the connection with our approach? If you clarify this for us, we will certainly be happy to add this reference in the camera-ready version of the manuscript.
> >
> >
> >
> > > **"I would have loved more experiments (and also more comparisons to comparable models)."**
> >
> > Thank you for this comment. We agree that the previous version of the manuscript seemed to lack in this regard. Therefore, we have added the following things:
> > 1. Additional performance comparison with VTA on the 3D Shapes and the Synthetic datasets -- Figures **3d** and **4d**;
> > 2. Additional comparison of VPR with VTA on a more challenging Bouncing Balls dataset -- Appendix **C.3**;
> > 3. Analysis of the MiniWorld Maze event detection (+ comparison with VTA) -- Appendix **C.6**;
> > 4. A more clear explanation of VPR's event prediction capabilities (+ comparison with CW-VAE) -- Section **4.4**.
> >
> > We hope these extra results provide a more compelling argument in favour of the proposed model.
> >
> > Furthermore, we have also added a new supplementary section (C.3 - Decomposing the KL terms of the CE criterion) discussing an interpretation of the CE criterion.
> >
> >
> > Thank you once again for your review and we really hope we could address your questions!

---

> > > ### Author Response · Authors · 2021-11-28
> > > **To Reviewer RjVA**
> > >
> > > Dear Reviewer RjVA,
> > >
> > > Thank you once again for your review of our work. As the discussion period is approaching its end, we would be grateful if you could confirm whether our responses and the additions we have made to the manuscript addressed your concerns, and let us know if any issues remain.

---

> > > > ### Author Response · Authors · 2021-11-30
> > > > **To Reviewer RjVA (2)**
> > > >
> > > > Dear Reviewer RjVA,
> > > >
> > > > The deadline for the discussion period is just a few hours away. We were therefore wondering if there is still any chance for you to let us know whether we have addressed your concerns with our response and the updates to the manuscript. We would very much appreciate it. Thank you!

---

### Official Review · Reviewer_xcsC · 2021-11-04

**Correctness:** 4
**Technical Novelty And Significance:** 3
**Empirical Novelty And Significance:** 3
**Recommendation:** 6
**Confidence:** 5

**Main Review:**

Strengths: In my knowledge, the proposed method of learning hierarchical temporal structures by measuring prior and posterior belief states is new in the area of unsupervised representation learning for video data.

Weaknesses: My major concern is about the insufficient experimental results.
- The model is compared with VTA only on the Moving Ball dataset, which in my view, is not enough to validate the superiority of the proposed model. The authors may consider to compare with VTA on a more challenging dataset such as Bouncing Balls. I believe this dataset can better support the claim in this paper, because it has more obvious event changes represented by the interactions of multiple balls and is also used in the work of VTA.
- The model is not compared with any existing approaches on the other three datasets.
- How does the rollout result in Fig 5, especially the color of the object, support the claim in the caption that "the produced rollouts illustrate model’s ability to...produce accurate...jumpy rollouts"?

Other concerns:
- How to determine the number of layers if we do not know how many latent factors (e.g., color, shape, floor...) there are in the environment?
- Although the authors claim in the appendix and the hyperparameters in Criterion U are robust across datasets, it is still not clear to me how to tune these hyperparameters. It would be good if the authors can give a sensitivity analysis on them.

**Summary Of The Paper:**

This paper presents a new model for unsupervised event detection, which has the following three technical contributions:
- It proposes individual criterion and mechanisms to detect expected and unexpected observations based on the comparison of prior and posterior latent representations. (Novel)
- It uses the above criterion to control the update rate of the latent states in a hierarchy of RNN layers. (Somewhat similar to VTA [Kim et al., 2019])
- The paper includes some interesting experiments to support the effectiveness of the proposed method. (But not convincing enough)



**Summary Of The Review:**

This paper explores an interesting problem of unsupervised representation learning in videos. Although the hierarchical architecture based on latent variables is very similar to the temporal abstraction structure in VTA [Kim et al., 2019], I think the proposed criteria for event detection are novel and reasonable. However, my major concern is about the insufficient experimental results (see my comments above), and so I give this paper a Weak Reject at this moment.

---

> ### Author Response · Authors · 2021-11-23
> **Response to Reviewer xcsC (Part 1)**
>
> Dear Reviewer xcsC,
>
> Thank you for your review!
>
> We are encouraged that you found our approach a novel and reasonable direction in unsupervised representation learning for videos. And we are also happy to hear that you liked our methods for analysing the effectiveness of the proposed model. In what follows, we wish to address some of the questions you raised.
>
>
> #### Main comments
>
>
> >"**The authors may consider to compare with VTA on a more challenging dataset such as Bouncing Balls. I believe this dataset can better support the claim in this paper, because it has more obvious event changes represented by the interactions of multiple balls and is also used in the work of VTA.**"
>
>
> Thank you for this suggestion! We agree that the single comparison was not enough. In order to validate our approach further, we have added three more comparisons -- VPR vs. VTA event detection on the *Synthetic*, *3D Shapes*, and *Bouncing Balls* (as you suggested) datasets. These results further demonstrate that VPR is more effective at the task of event detection. We hope we could address your concern with this.
>
>
>
> >"**The model is not compared with any existing approaches on the other three datasets**"
>
> As mentioned in the previous comment, we have added three additional comparisons.
>
>
> > "**How does the rollout result in Fig 5, especially the color of the object, support the claim in the caption that "the produced rollouts illustrate model’s ability to...produce accurate...jumpy rollouts"?**"
>
> This is a very important question that we wish to explain more clearly. Figure 5 demonstrates VPR rollouts performed with respect to each of its latent levels. As explained in the paper, each of the levels learns to represent an appropriate factor of variation. For *Moving Ball*: Level 1 represents the position of a ball and thus learns to predict the position in the future; Level 2 represents the colour of a ball and thus learns to predict the colours in the future. Same logic applies to the 3D Shapes dataset.
>
> As discussed, this is primarily driven by the event detection mechanism that  matches the rate at which a particular feature changes over time with the rate at which the corresponding level updates its state. By consequence, this implies that VPR learns to transition these features from one event boundary to the next -- which is what is demonstrated in Figure 5. Although the colour of the ball changes every *few* steps, VPR learns to transition it at *every* *subjective* timestep of its rollout. For instance, although the Moving Ball (Level 2) rollout you see in Figure 5 is 17 steps, in a physical timesale this translates to about 64 steps (since ball colour changes approximately every 4 steps). This is why we called it a "*feature-specific and jumpy rollout*".
>
> Further, it is "accurate" since VPR predicts the correct order of colour changes (see the ground-truth rollout). This was also demonstrated using the 3D Shapes event prediction accuracy (Section **4.2** - paragraph 2). To elucidate this property more, we have added a separate short section (**4.4**) dedicated to the reporting of event prediction accuracy. We hope we could answer your question.

---

> > ### Author Response · Authors · 2021-11-23
> > **Response to Reviewer xcsC (Part 2)**
> >
> > >"**Although the hierarchical architecture based on latent variables is very similar to the temporal abstraction structure in VTA [Kim et al., 2019], I think the proposed criteria for event detection are novel and reasonable.**"
> >
> > Thank you for your positive assessment of our model! Here, we would like to briefly comment on the assumption that VPR's and VTA's latent architectures are similar. At present form, VTA is a two-level hierarchical generative model with an explicit parametrised boundary detection mechanism. While it may be possible to extend VTA's architecture to a higher number of hierarchical levels (including its detection mechanism), the task does not seem to be an easy one and was not studied in the original paper. Hence, the reason why we thought it is important to follow-up on this comment: it is one of the contributions of our paper to generalise a hierarchical generative model with flexible (and nested, in our case) timescales to an arbitrary number of layers. We very much appreciate that you acknowledge the novelty of our non-parametric event detection technique, but we wanted to mention that its novelty and effectiveness are inseparably linked to the devised generative model with nested timescales.
> >
> > Furthermore, although the graphical models of VTA and two-level VPR may seem similar, there is an important distinction. Specifically, the generated reconstructions in VPR are conditioned on all hierarchical latent states, $p(o_t|s_t^{1:N})$, similar to NVAE [Vahdat and Kautz, 2020]. This allows our model to effectively combine information represented in the different levels and produce diverse reconstructions and rollouts (which we also hypothesise improves disentanglement properties). This property is more clearly discussed in Appendix **C.1**, where we block VPR's temporal information when producing rollouts. This contrasts with methods like CW-VAE and VTA, which produce reconstructions using the bottom-level latent state, $p(o_t|s_t^1)$.
> >
> >
> >
> > #### Other minor comments:
> >
> > >"**How to determine the number of layers if we do not know how many latent factors (e.g., color, shape, floor...) there are in the environment?**"
> >
> > This is a brilliant question. This problem is common and seems somewhat analogous to selecting the number of centroids in a k-means algorithm or latent dimensions in a $\beta$-VAE. Although we leave this for future work, at present we hypothesise that, similar to the behaviour in an overspecified latent space of a VAE, the higher levels of VPR's hierarchy will collapse and cease to represent anything if deemed superfluous for accurate reconstructions.
> >
> >
> > >"**It would be good if the authors can give a sensitivity analysis on the [hyperparameters of the Criterion U].**"
> >
> > This is a great request. We have added a sensitivity analysis section in Appendix **C.4**.
> >
> >
> >
> > Thank you again for your review! We really hope we could address your concerns and are looking forward to hearing back from you.

---

> > > ### Comment · Reviewer_xcsC · 2021-11-25
> > > **Waiting for more explanations.**
> > >
> > > I appreciate the detailed explanations and the new results. I do have two additional minor questions.
> > > 1. For the Bouncing Ball dataset, in Figure 13, how did you determine the labels of the event boundaries? By position or by color?
> > > 2. Can the model detect other more dynamics-related events, such as interactions?

---

> > > > ### Author Response · Authors · 2021-11-25
> > > > **Response to Reviewer xcsC**
> > > >
> > > > Dear Reviewer xcsC,
> > > >
> > > > We are happy to clarify these two questions.
> > > >
> > > > > **"For the Bouncing Ball dataset, in Figure 13, how did you determine the labels of the event boundaries? By position or by color?"**
> > > >
> > > > For the Bouncing Balls dataset (just like in the Moving Ball dataset), we consider two temporal factors of variation -- position (fast) and colour (slow). As in the VTA [Kim et al., 2019] paper, the ball changes its direction and colour upon hitting a wall or upon bouncing off another ball (interaction). In both datasets, a change in *direction/colour* is labeled as an event.
> > > >
> > > >
> > > > > **"Can the model detect other more dynamics-related events, such as interactions?"**
> > > >
> > > > Indeed, from our experiments, ball interactions are actually the most accurately detected category of events, achieving a true-positive rate of 96.7% ($\pm$ 2.9%). However, note that there are no dynamics-related interactions that are not associated with a colour change -- this property is reproduced from the VTA [Kim et al., 2019] paper dataset.

---

> > > > > ### Author Response · Authors · 2021-11-28
> > > > > **To Reviewer xcsC**
> > > > >
> > > > > Dear Reviewer xcsC,
> > > > >
> > > > > Thank you once again for your review of our work. As the discussion period is approaching its end, we would be grateful if you could confirm whether our responses and the additions we have made to the manuscript addressed your concerns, and let us know if any issues remain.

---

> > > > > ### Comment · Reviewer_xcsC · 2021-11-29
> > > > > **Increased my rating by 1**
> > > > >
> > > > > The authors added new experiments on the requested dataset and sufficiently clarified the experimental configurations of my concern, and so I decided to increase my rating by 1, even though I was not as impressed by the novelty of the proposed model (compared to VTA). I strongly encourage the authors to include more comparisons with VTA in the main body of the revised paper.

---

> > > > > > ### Author Response · Authors · 2021-11-29
> > > > > > **Response to Reviewer xcsC after increasing the rating**
> > > > > >
> > > > > > Dear Reviewer xcsC,
> > > > > >
> > > > > > > **"The authors added new experiments on the requested dataset and sufficiently clarified the experimental configurations of my concern, and so I decided to increase my rating by 1"**
> > > > > >
> > > > > > We are encouraged to hear you found our explanations sufficient -- thank you for increasing your score!
> > > > > >
> > > > > > > **"[...] even though I was not as impressed by the novelty of the proposed model (compared to VTA)."**
> > > > > >
> > > > > > With respect to the similarity of our model to VTA, we are not sure what you are referring to since it wasn't clarified in your review. Still, we wish to reiterate the main points that show distinct differences between the two models:
> > > > > >
> > > > > > 1. VPR uses an entirely different *boundary detection mechanism* compared to VTA (non-parametric vs. parametric) which significantly outperforms VTA in detecting events (see Figures 3b-d, 4d, 13);
> > > > > > 2. VPR is a generative model with an *arbitrary* number of hierarchical levels compared to VTA with only 2 levels; it is unclear how to effectively scale VTA and its detection mechanism;
> > > > > > 3. VPR can employ *several* boundary detection mechanisms, which is afforded by the scalability of VPR to an arbitrary number of layers and its non-parametric detection system; we showed that VPR is therefore capable of detecting different kinds of feature-specific events with high accuracy (using 3D Shapes dataset in Figure 3c);
> > > > > > 4. VPR's and VTA's graphical models are *different* in both the reconstruction and prior/posterior models, mostly due to point 2. These represent significant architectural differences, since it necessitates the integration of deterministic variables, as shown in Figures 1 and 2:
> > > > > >     - VPR reconstructions are conditioned on all hierarchical latent states $p(o_t|s^{1:N}_t)$ vs. just the bottom state $p(o_t|s^1_t)$ in VTA;
> > > > > >     - VPR priors and posteriors are conditioned on all hierarchical states above $s^{>n}$ (realised using deterministic variable $c$) and level-specific observation embeddings $x^{n}$ (realised using deterministic variable $x$);
> > > > > >
> > > > > > Nevertheless, if the lack of novelty of VPR is considered solely because of the assumed similarity of the (*two-level*) VPR's graphical model with that of VTA (please see point 4 for the architectural differences), we believe this may be a harmful perspective for future research in this area. It seems that any alternative architecture which employs a hierarchy of latent states updating at different rates (even though VTA has only two levels) will be deemed not novel by this logic (e.g. CW-VAE [1]). We hope you could reconsider your opinion on this subject, or provide an argument for why you believe the differences are not substantial. Thank you!
> > > > > >
> > > > > > [1] Saxena et al., NeurIPS, 2021

---

### Author Response · Authors · 2021-11-23
**To all Reviewers**

Dear Reviewers,

We thank you very much for your feedback on our work. We are happy to see that you acknowledged the novelty and the demonstrated effectiveness of our proposed model in extracting and learning disentangled spatiotemporal features from video sequences.


Although we have replied to each of you individually, we would like to reiterate certain points of confusion and summarise the updates we have made to the manuscript.


**Overview**

Our paper explores a generative model architecture that uses a non-parametric event detection mechanism to enforce a hierarchy of nested timescales. The combination of nested and subjective timescales, hierarchical structure, and the event detection mechanism result in the VPR's ability to learn disentangled spatiotemporal representations across the hierarchy of the model. We were happy to see that the reviewers were in joint agreement that this key property of VPR was evident from our experiments and analysis.

The event detection mechanism employs VPR's generative model to compute information-theoretic quantities in order to identify changes (i.e. events). The novelty of this event detection is similarly acknowledged by the reviewers and includes two main points:

(1) it is non-parametric, and thus a separate parametrised model is not required to be trained (in contrast to the previous works reviewed in the paper);
(2) it is present at every level of VPR's latent hierarchy, and thus detects events with respect to the particular temporal factors of variation present in the corresponding levels of VPR.

In its turn, the detection mechanism determines the rates at which each of the VPR's hierarchical levels update over time -- these rates correspond to the rates at which the represented features change over time. Because of this, VPR learns to transition between event boundaries directly, thus performing *feature-specific* and *jumpy* rollouts.


**Summary of the updates**

With respect to the questions and suggestions you provided, we would like summarise the changes we have made to the manuscript:
1. Additional performance comparison with VTA on the 3D Shapes and the Synthetic datasets -- Figures **3d** and **4d**.
2. Additional comparison of VPR with VTA on a more challenging Bouncing Balls dataset -- Appendix **C.3**.
3. Analysis of the MiniWorld Maze event detection (+ comparison with VTA) -- Appendix **C.6**.
4. A more clear explanation of VPR's event prediction capabilities (+ comparison with CW-VAE) -- Section **4.4**.
5. Minor additions include:
    * Sensitivity analysis of the CU criterion hyperparameters -- Appendix **C.4**.
    * Pseudocode -- Appendix **B.1**.
    * More insight into the CE criterion -- Discussion and Appendix **C.5**.
    * An additional citation [Schmidhuber, 1991].


We thank you once again for your reviews and sincerely hope we could address your questions!



========== EDIT ==========

Thank you to all of the reviewers who have responded. We are happy to hear our aforementioned updates and further clarifications reassured the reviewers, as they have increased their scores accordingly.

---

### Decision · Program_Chairs · 2022-01-20

**Decision:**

Accept (Poster)

**Comment:**

Thanks for your submission to ICLR.

This paper considers a variational inference hierarchical model called Variational Predictive Routing.

Prior to discussion, several reviewers were on the fence about the paper, most notably having concerns about some of the experimental results as well as various clarity issues throughout the paper.  However, the authors did a really nice job addressing many of these concerns.  Ultimately, several of the reviewers updated their scores, leading to a clear consensus view that this paper is ready for publication.  We really appreciate your effort in providing additional details and results.

Please do keep in mind the concerns of the reviewers when preparing a final version of the manuscript.